# Temporal-aware Flow Matching for Video Generation with Temporally Coherent Motion

**Zirui Pan** [1]  **Xin Wang** [1]  **Yipeng Zhang** [1]  **Yuwei Zhou** [1]  **Wenwu Zhu** [1]

## Abstract

Despite rapid advances in text-to-video generation, state-of-the-art generative models still suffer from producing temporally incoherent and unrealistic motion for videos. The key weakness of existing works is that they commonly treat videos as frame sequences and directly adopt Flow Matching (FM) objectives, which are originally designed for images. This practice fails to explicitly model motion priors or temporal dependencies, resulting in suboptimal dynamics that may appear incoherent and unrealistic. To solve this problem, we propose Temporal-aware Flow Matching (TFM), a novel training paradigm that embeds inter-frame constraints into the flow objective, leading to temporally coherent motion modeling in video generation. More specifically, the proposed TFM enforces temporal correlations across frames while retaining the desirable properties of FM, and further introduces a residual-type loss that aligns naturally with this new flow. We theoretically prove that models trained with TFM are able to exhibit remarkably enhanced temporal perception ability. Notably, TFM imposes no additional cost during inference and is applicable to any model using FM. Extensive experiments demonstrate that our TFM can significantly improve motion realism across diverse motion types. Generated videos are presented at https://pzrain.github.io/tfm.

## 1. Introduction

Despite rapid progress in text-to-video generation (Brooks et al., 2024; Esser et al., 2024; Kodaira et al., 2025), existing

---

[1]Department of Computer Science and Technology, BNRist, Tsinghua University, Beijing, China. Correspondence to: Xin Wang <xin_wang@tsinghua.edu.cn>, Wenwu Zhu <wwzhu@tsinghua.edu.cn>.

*Proceedings of the $43^{rd}$ International Conference on Machine Learning*, Seoul, South Korea. PMLR 306, 2026. Copyright 2026 by the author(s).

state-of-the-art approaches, even those billion-parameter models (Wan et al., 2025; Wu et al., 2025) capable of producing visually high-quality clips, still suffer from a critical problem, *i.e.*, how to generate temporally coherent and realistic motion (Hu et al., 2025) in videos. This problem is largely caused by two factors, *i.e.*, (i) the violation of basic physical laws (*e.g.*, Figure 1(a) demonstrates the failure of the baseline in rendering water splashes when feet step on the ground), and (ii) incoherent or unstable scenes under complex motion (*e.g.*, Figure 1(b) shows that the sample generated by baseline displays visible body deformations).

In this paper, we study the problem of video generation with temporally coherent motion, which poses a fundamental challenge in current video models, *i.e.*, the *inability* to effectively internalize motion priors from video data. Although motion naturally arises from the temporal relationships between adjacent frames and should, in principle, be learnable directly from large video corpora, empirical evidence suggests that this remains challenging in practice (Kang et al., 2025). Even when trained on massive video datasets, it is extremely difficult to capture coherent and physically consistent motion, thus resulting in dynamics that violate physical constraints or become unstable under complex scenarios.

To tackle this fundamental challenge, we first provide an unprecedented perspective to understand why video generative models struggle to learn motion priors and then propose a new paradigm for training general video generators. We argue that the root cause lies in the prevailing training objective used in current generative models—Flow Matching (FM) (Lipman et al., 2023; Liu et al., 2023a). Although effective for image generation, FM does not **explicitly** encode motion priors. As a result, video models are forced to rely on **implicit** attention mechanisms to infer temporal structure, which is fundamentally suboptimal for modeling videos. Originally designed for image synthesis, FM learns a **direct** probability path (flow) from noise to data. Since this framework is theoretically sound and empirically effective in the image domain, current video models that largely treat videos as sequences of images have directly extended it to video generation. However, these approaches typically reuse the original FM objective without necessarily conducting video-specific modifications (Davtyan et al., 2023;

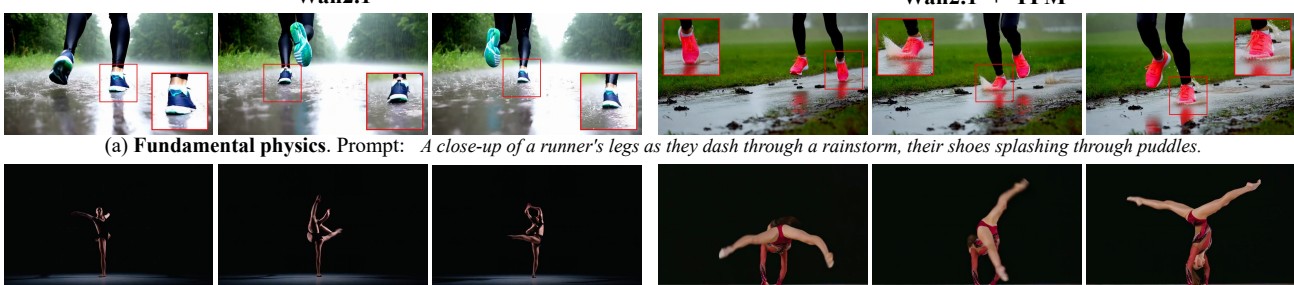

*Figure 1.* Typical examples of motion incoherence generated by the base model Wan2.1-T2V-14B (Wan et al., 2025), where the model fails to produce realistic dynamics, *e.g.*, no water splashes when stepping on the ground in **(a) Fundamental Physics**, and noticeable body deformation in **(b) Complex Motion**. In contrast, Wan2.1 equipped with TFM produces videos with coherent and text-faithful motions.

Lin et al., 2024), leaving the flow of each frame temporally independent and failing to capture the inherent temporal correlations that manifest as motion. Consequently, models trained in this way are prone to producing incoherent or physically implausible dynamics.

Motivated by these observations, we propose a novel training paradigm, *i.e.*, Temporal-aware Flow Matching (TFM), to enhance motion modeling in video generation. Conceptually, TFM overcomes the limitations of the direct yet temporally decoupled flows in standard FM by introducing a new probability path that embeds motion priors into the flow itself. Specifically, in Section 3.2, we reform the original FM governing equation (Eq (3), which only enforces intra-frame constraints) to incorporate inter-frame constraints (Eq (6)). Compared to the flow in FM, the TFM flow introduces a training objective that explicitly captures the temporal correlations inherent across frames, making it better suited for temporally structured data such as videos. To optimize the model under this new flow, we propose a residual-type loss in Section 3.3, which is shown to be more effective than the solution-type loss used in FM. In Section 4, we further establish a theoretical foundation for TFM, proving that models trained under TFM exhibit enhanced temporal perception and are able to learn motion dynamics much better than those trained with FM. Importantly, TFM introduces no additional time or memory overhead during inference and is applicable to any video generative model that utilizes the flow-based training objective. Extensive experiments show that TFM significantly improves temporal coherence across diverse motion types for video generative models. To summarize, we make the following contributions:

- We propose a novel training paradigm, *i.e.*, Temporal-aware Flow Matching (TFM), to enhance motion modeling in video generation. Being superior to FM, TFM explicitly captures temporal correlations, enabling models to effectively learn motion priors.

- We provide a theoretical foundation for TFM, proving that models trained under TFM yield stronger temporal perception compared to those trained with FM.

- We empirically validate the effectiveness of the proposed TFM through extensive experiments, demonstrating significant improvements in motion coherence.

## 2. Related Work

**Flow-based Generative Models** Diffusion models (Ho et al., 2022; Blattmann et al., 2023b; Rombach et al., 2022) have revolutionized video generation with their exceptional generative capabilities. Early works extended DDPM (Ho et al., 2020) to text- or image-to-video tasks, modeling temporal dynamics primarily through implicit attention mechanisms (Guo et al., 2023; Luo et al., 2023; Zhang et al., 2025a; Blattmann et al., 2023a; Chen et al., 2025; Zhang et al., 2025b). More recently, Flow Matching (Lipman et al., 2023; Liu et al., 2023a) reformulates generation as a distribution-mapping problem, providing a more robust and theoretically grounded training objective. It has been further generalized to general geometries (Chen & Lipman, 2024), discrete data (Gat et al., 2024) and high-order velocity fields (Su et al., 2025), establishing FM as the SoTA training framework for generative tasks.

Nevertheless, existing video generative models based on flow matching (HaCohen et al., 2024; Liu et al., 2025; Wang et al., 2025c; Ma et al., 2024; Wu et al., 2025; Liu et al., 2023b) remain fundamentally similar to DDPM-based approaches: they model videos as sequences of frames and learn frame-wise, direct yet independent probability paths. By constructing flows separately for each frame, these methods fail to explicitly capture temporal dependencies or motion priors, often leading to physically implausible or inconsistent motion, especially in scenes with complex dynamics or interactions (Kang et al., 2025), limiting the models' ability to produce temporally coherent video sequences.

**Refined Motion in Generation** To address the universal challenge of generating realistic motion, prior works have explored two main directions: (1) Physics-informed approaches assume that motion is strictly governed by physical laws. They incorporate physics-based constraints into

diffusion models, enforcing the generated distribution to satisfy underlying partial differential equations (Bastek et al., 2024). However, such methods typically operate on highly specialized domains, such as atmospheric systems (Wang et al., 2025b) or turbulent flows (Shu et al., 2023), and can hardly generalize to real-world videos, where multiple interacting physical processes coexist; (2) Externally guided approaches share our view of motion as temporal structure, and introduce additional signals as guidance. Early works leverage large language models to plan scene layouts (Lian et al., 2024; Huang et al., 2023; Hong et al., 2023), offering coarse guidance such as object movement or prompt refinement (He et al., 2025). More recent methods adopt fine-grained control signals, including optical flow (Liang et al., 2024; Nam et al., 2025; Chefer et al., 2025) and point clouds (Ren et al., 2025; LAN et al., 2025; Wang et al., 2025a), either as conditioning inputs or jointly learned cues. While effective, these strategies rely on external supervision, suggesting that existing models fail to fully exploit the motion information already inherent in video data. We argue that a generative model should be able to learn such motion priors directly from data without additional signals, provided that the training objective is properly designed.

## 3. Method

In this section, we detail our method. We first review standard FM and explain why its direct application to video fails to explicitly model temporal dynamics. We then introduce TFM and describe the key implementation details required for practical use. Finally, we present intuitive examples that illustrate the temporal constraints imposed by our TFM.

### 3.1. Preliminaries

Flow Matching (Lipman et al., 2023; Liu et al., 2023a) is a methodology widely adopted in recent state-of-the-art generative models. It formulates generation as a transportation problem between probability distributions: the model learns a mapping that transports samples from a simple prior distribution, *i.e.*, standard Gaussian, to the unknown data distribution. This is accomplished by fitting a probability path, *i.e.*, *flow*, which in standard FM corresponds to a straight path between the two distributions. Specifically in training, given data $x_1 \in \mathbb{R}^d$, random noise $x_0 \in \mathbb{R}^d \sim \mathcal{N}(0, I)$ and a timestep $t \in [0, 1]$, the noised intermediate latent $x_t$ and the corresponding constant velocity $u_t$ is calculated as:

$$x_t = tx_1 + (1-t)x_0, \quad u_t = x_1 - x_0, \quad (1)$$

which forces the flow between the two distributions to be a direct line in latent space. The model is then optimized by:

$$\mathcal{J} = \mathbb{E}_{x_1,x_0 \sim \mathcal{N}(0,1), t \in [0,1]}[||u_\theta(x_t, t) - u_t||_2^2], \quad (2)$$

where $u_\theta(x_t, t)$ denotes the model's predicted velocity, and $\theta \in \mathcal{M}$ represents the learnable parameters. In its subse-

quent work (Chen & Lipman, 2024), the flow, denoted as $\psi_t(x|x_1)$, is further generalized to be a solution to Eq (3):

$$d(\psi_t(x|x_1), x_1) = \tau(t)d(x_0, x_1), \quad (3)$$

where $\tau(t)$ is a monotonically decreasing function satisfying $\tau(0) = 1$ and $\tau(1) = 0$, and $d(\cdot, \cdot) : \mathbb{R}^d \times \mathbb{R}^d \to \mathbb{R}^d$ is a metric function satisfying the following constraints:

1. *Non-negative*: $d(x, y) \geq 0$.
2. *Positive*: $d(x, y) = 0$ iff $x = y$.
3. *Non-degenerate*: $\nabla d(x, y) \neq 0$ iff $x \neq y$.

These constraints ensure that the flow reaches its target at $t = 1$, *i.e.*, $\psi_1(x|x_1) = x_1$. When setting $\tau(t) = 1 - t$, $d(\cdot, \cdot)$ the Euclidean distance, and constraining the transport direction to be $x_1 - x_0$, the solution to Eq (3) collapses precisely to the straight-constant flow used in standard FM.

Building on this formulation, we now consider its extension to video generation. Suppose we have a video consisting of $n$ frames $x_1 = \{x_1^{(1)}, x_1^{(2)}, \cdots, x_1^{(n)}\}$, where each frame $x_1^{(i)} \in \mathbb{R}^{c \times h \times w}$ is paired with a noise $x_0^{(i)}$ of the same shape, $1 \leq i \leq n$. While FM was originally proposed for images, video data introduce temporal structure, a further dimension of complexity. A video is not merely a collection of images but an ordered sequence with dependencies across time. Nevertheless, existing FM-based video models (Wan et al., 2025; Wu et al., 2025) commonly simplify this structure by treating all frames as independent and constructing a separate flow for each of them. Concretely, they define

$$x_t^{(i)} = tx_1^{(i)} + (1-t)x_0^{(i)}, \quad 1 \leq i \leq n, t \in [0, 1], \quad (4)$$

as a linear interpolation between noise and data, and

$$u_t^{(i)} = x_1^{(i)} - x_0^{(i)}, \quad 1 \leq i \leq n, \quad (5)$$

as the velocity, enforcing a frame-wise straight-constant path. Such a setup fails to explicitly model the temporal relationships across frames, which we hypothesize may impair the model's ability to capture motion priors inherent in video data and generate coherent dynamics.

### 3.2. Temporal-aware Flow Matching

Motivated by our hypothesis that standard FM fails to explicitly capture the temporal dynamics inherent in videos, we propose a novel training paradigm Temporal-aware Flow Matching (TFM). TFM extends the basic FM governing equation (Eq (3)) to the following system of equations:

$$\begin{aligned}
&\rho \cdot d(\psi_t(x|x_1^{(i)}), x_1^{(i)}) + \mathbb{I}_{i>1}d(\psi_t(x|x_1^{(i-1)}), \psi_t(x|x_1^{(i)})) \\
&+ \mathbb{I}_{i<n}d(\psi_t(x|x_1^{(i)}), \psi_t(x|x_1^{(i+1)})) \\
=&\rho \cdot \tau(t)d(x_0^{(i)}, x_1^{(i)}) \\
&+ \mathbb{I}_{i>1}[\tau(t)d(x_0^{(i-1)}, x_0^{(i)}) + (1-\tau(t))d(x_1^{(i-1)}, x_1^{(i)})] \\
&+ \mathbb{I}_{i<n}[\tau(t)d(x_0^{(i)}, x_0^{(i+1)}) + (1-\tau(t))d(x_1^{(i)}, x_1^{(i+1)})],
\end{aligned}$$
(6)

where $1 \leq i \leq n$. This formulation consists of three components. The first term corresponds to the standard FM objective, with a hyper-parameter $\rho$ controlling its relative strength. The remaining two terms (highlighted in red and blue) are temporal terms that explicitly encode dependencies between adjacent frames: the current frame $x_1^{(i)}$ and its preceding frame $x_1^{(i-1)}$ (red), as well as its succeeding frame $x_1^{(i+1)}$ (blue). Here, $\mathbb{I}$ is the indicator function. These temporal terms are intentionally kept in their simplest form to capture the inherent adjacency structure of video data without adding excessive complexity. A corresponding interpolation term is included on the right-hand side to maintain consistency of the overall formulation. The constraints on $\tau(t)$ follow those of standard FM. For the metric function $d(\cdot, \cdot)$, we additionally impose the following requirement:

4. *Triangle Inequality*: $d(x, y) + d(y, z) \geq d(x, z)$.

Before solving the system to derive the corresponding velocity field, we first establish the validity of our formulation. We state the following proposition:

**Proposition 3.1.** *Eq* (6) *($\rho \geq 4$) characterizes flows that reach $x_1^{(i)}$ at $t = 1$, for all $1 \leq i \leq n$.*

The full proofs for our theorems are all provided in Appendix A. To streamline the discussion below, we introduce several additional notations:

$$
\begin{cases}
A_i = \nabla_x d(\psi_t(x|x_1^{(i)}), x_1^{(i)}), \ 1 \leq i \leq n, \text{ (intra-frame } i) \\
B_{i,j} = \nabla_x d(\psi_t(x|x_1^{(i)}), \psi_t(x|x_1^{(j)})) \\ \quad\quad\quad\quad\quad\quad\quad\quad\quad 1 \leq i < j \leq n, \\
B_{j,i} = \nabla_y d(\psi_t(x|x_1^{(i)}), \psi_t(x|x_1^{(j)})) \text{ (inter-frame } i \leftrightarrow j) \\
C_i = \frac{d}{dt}\tau(t) \begin{bmatrix} \rho d(x_0^{(i)}, x_1^{(i)}) + \mathbb{I}_{i>1}(d(x_0^{(i-1)}, x_0^{(i)}) - d(x_1^{(i-1)}, x_1^{(i)})) \\ + \mathbb{I}_{i<n}(d(x_0^{(i)}, x_0^{(i+1)}) - d(x_1^{(i)}, x_1^{(i+1)})), 1 \leq i \leq n \end{bmatrix},
\end{cases}
$$
(7)

We then present the following theorem:

**Theorem 3.2.** $u_t(x) = \begin{bmatrix} u(\psi_t(x|x_1^{(1)})) & \cdots & u(\psi_t(x|x_1^{(n)})) \end{bmatrix}^T$ *generates the flow defined in Eq* (6), *if and only if it satisfies the following ODE system $M u_t(x) = C$, where*

$$
M = \begin{bmatrix}
\rho A_1^T & B_{2,1}^T & \cdots & & & \\
+B_{1,2}^T & & & & & \\
& \ddots & & & & \\
& & \rho A_i^T & & & \\
\cdots & B_{i-1,i}^T & +B_{i,i-1}^T & B_{i+1,i}^T & \cdots & \\
& & +B_{i,i+1}^T & & & \\
& & & \ddots & & \\
& & \cdots & B_{n-1,n}^T & \rho A_n^T & \\
& & & & +B_{n,n-1}^T &
\end{bmatrix},
$$
(8)

*and $C = [C_1, \cdots, C_n]^T$.*

The matrix $M$ derived in Theorem 3.2 takes the form of an $n \times n$ block tridiagonal matrix, where each block corresponds to the respective derivatives appearing in Eq (6). For practical implementation, we instantiate the components

with commonly used choices in order to keep the resulting regression problem simple. Specifically, we set $\tau(t) = 1 - t$, use the Euclidean distance for $d(\cdot, \cdot)$, and constrain the direction of each $u(\psi_t(x|x_1^{(i)}))$ to align with $x_1^{(i)} - x_0^{(i)}$, matching the choices in standard FM. Note, however, that these constraints lead to solutions fundamentally distinct from those obtained in FM. Under these choices, all $A_i$, $B_{i,j}$ and $B_{j,i}$ in Eq (7) collapse to unit vectors:

$$
\begin{cases}
A_i = \widehat{\psi_t(x|x_1^{(i)}) - x_1^{(i)}} \triangleq -\hat{v}^{(i)}, \\
B_{i,j} = \widehat{\psi_t(x|x_1^{(i)}) - \psi_t(x|x_1^{(j)})} \triangleq \hat{v}^{(i,j)}, \\
B_{j,i} = \widehat{\psi_t(x|x_1^{(j)}) - \psi_t(x|x_1^{(i)})} \triangleq \hat{v}^{(j,i)},
\end{cases}
$$
(9)

where $\hat{z}$ denotes the unit vector in the direction of any vector $z$, and note that $\hat{v}^{(i)} = \widehat{x_1^{(i)} - x_0^{(i)}}$ is the direction of the straight path from noise to data. We can therefore define:

$$
\bar{u}_t(x) = \begin{bmatrix} u(\psi_t(x|x_1^{(1)}))^T \hat{v}^{(1)} & \cdots & u(\psi_t(x|x_1^{(n)}))^T \hat{v}^{(n)} \end{bmatrix}^T,
$$
(10)

which captures the per-frame velocity magnitudes, as the directions have been pre-determined. The resulting system then reduces to a linear system $\bar{M}\bar{u}_t(x) = C$, where

$$
\begin{cases}
\bar{M}_{i,i-1} = \hat{v}^{(i-1,i)T}\hat{v}^{(i-1)}, i > 1, \\
\bar{M}_{i,i} = \mathbb{I}_{i>1}\hat{v}^{(i,i-1)T}\hat{v}^{(i)} + \mathbb{I}_{i<n}\hat{v}^{(i,i+1)T}\hat{v}^{(i)} - \rho, 1 \leq i \leq n, \\
\bar{M}_{i,i+1} = \hat{v}^{(i+1,i)T}\hat{v}^{(i+1)}, i < n, \\
C_i = -\rho d(x_0^{(i)}, x_1^{(i)}) - \mathbb{I}_{i>1}(d(x_0^{(i-1)}, x_0^{(i)}) - d(x_1^{(i-1)}, x_1^{(i)})) \\
\quad\quad - \mathbb{I}_{i<n}(d(x_0^{(i)}, x_0^{(i+1)}) - d(x_1^{(i)}, x_1^{(i+1)})), 1 \leq i \leq n.
\end{cases}
$$
(11)

We set $\rho = 4.0$ which ensures that $\bar{M} \in \mathbb{R}^{n \times n}$ is diagonally dominant. Under this condition, the system can be efficiently solved using the Thomas algorithm (Thomas, 1949), which is both fast and numerically stable.

In summary, we construct a temporally constrained flow across video frames that differs fundamentally from the standard FM formulation. The detailed reasoning behind the derivation of TFM (Eq (6)) is provided in Appendix D.1. Although the resulting overall transformations between the source and target distributions are mathematically equivalent, we argue that our flow is more conducive for generative models to learn motion priors, namely the fundamental temporal dependencies present in video data.

### 3.3. Implementation

In this section, we discuss the practical implementation of TFM, covering both training and inference. During inference, TFM introduces no change to the standard pipeline and does not add any time or memory overhead. During training, although the overall *v-prediction* framework remains intact, two key modifications are required.

**Solving the system** Training under the v-prediction paradigm requires access to the latent state at timestep $t$ and

its corresponding ground-truth velocity field. While Theorem 3.2 allows us to efficiently compute each $u(\psi_t(x|x_1^{(i)}))$ given the latents $\psi_t(x|x_1^{(i)})$, the latents themselves do not admit a closed-form expression. In fact, except for special cases, such as the straight-constant flow in FM or certain interpolation-based curves, most non-trivial flows lack analytical solutions. Consequently, the latent trajectories must be obtained by numerically integrating the ODE system.

In practice, we find that a simple Euler discretization suffices to produce accurate latent trajectories. Moreover, the additional computational overhead incurred by this integration step is marginal relative to the overall optimization cost. Further analysis is provided in Appendix D.3.

**Optimizing the model**  We disentangle the target velocity into magnitude and direction, and define the training objectives respectively:

$$J(\theta) = \mathbb{E}_{t,x_1,x_0}[||\bar{M}\bar{u}_{\theta,t}(x) - C||_2^2] -$$
$$\mathbb{E}_{t,x_1,x_0,1\leq i\leq n}[\text{cos\_sim}(\widehat{u_\theta(\psi_t(x|x_1^{(i)}))}, \widehat{x_1^{(i)} - x_0^{(i)}})]. \tag{12}$$

For the direction term, we use cosine similarity, and for the magnitude term we adopt a **residual-type** loss, denoted as $\text{loss}^{\text{TFM}} = \mathbb{E}||\bar{M}\bar{u}_{\theta,t}(x) - C||_2^2$. An alternative is to solve the system to obtain the unique solution $\bar{u}_t^*(x)$, as in standard FM, and define a **solution-type** loss $\text{loss}^{\text{FM}} = \mathbb{E}||\bar{u}_{\theta,t}(x) - \bar{u}_t^*(x)||_2^2$. We next explain why $\text{loss}^{\text{TFM}}$ is more effective compared to $\text{loss}^{\text{FM}}$ in our setting.

For a well-trained model, the predicted velocity should closely follow the probability path. To measure deviations, we define the *equation-residual indicator*:

$$\Lambda(u) = \mathbb{E}||\bar{M}u - C||_2^2. \tag{13}$$

A smaller $\Lambda(u)$ indicates better adherence to the TFM equations. Our loss $\text{loss}^{\text{TFM}}$ is constructed to directly minimize $\Lambda$. Although both losses share the same global minimum, they differ in how they *rank* intermediate predictions. To clarify this, we introduce the notion:

**Definition 3.3.** (*Consistent objective*) Let $L(\theta)$ be a training objective for model parameters $\theta \in \mathcal{M}$. We call $L$ a *consistent objective* (with respect to $\Lambda$) if and only if

$$L(\theta_1) \leq L(\theta_2) \Leftrightarrow \Lambda(\bar{u}_{\theta_1}) \leq \Lambda(\bar{u}_{\theta_2}), \tag{14}$$

for all parameter choices $\theta_1, \theta_2 \in \mathcal{M}$.

In other words, a *consistent objective* must produce the same ranking of model parameters as $\Lambda$. By design, $\text{loss}^{\text{TFM}}$ is consistent. In contrast, the following proposition holds:

**Proposition 3.4.** $\text{loss}^{\text{FM}}$ *is **not** a consistent objective.*

Proposition 3.4 implies that $\text{loss}^{\text{FM}}$ may assign equal penalties to two predictions even if one satisfies the TFM equations substantially better. Thus, although both losses share

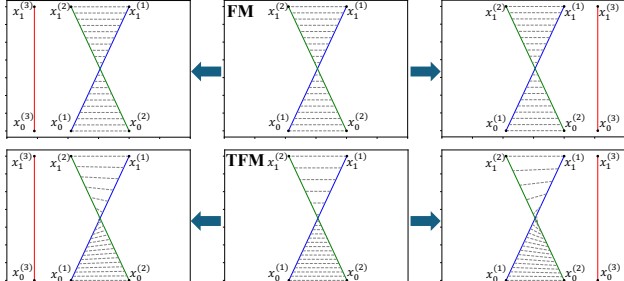

*Figure 2.* Illustration of flow interactions (dashed lines) under FM and TFM using simple 2D data. The middle subfigures show only two frames. In the left and right subfigures, we add and shift a third frame, observing how the flow changes accordingly.

a global minimum, only $\text{loss}^{\text{TFM}}$ aligns with the probability path principle central to TFM. In the standard FM setting, the system reduces to $n$ independent equations, and $\text{loss}^{\text{TFM}}$ will degenerate to the standard $\text{loss}^{\text{FM}}$.

### 3.4. Discussion

In this section, we discuss the connection between TFM and standard FM, together with the numerical stability of the proposed optimization process. First, to provide an intuitive comparison between FM and TFM, we visualize their induced probability paths on simple 2D data in Figure 2. In the middle subfigures, we consider two flows, $x_0^{(1)} \to x_1^{(1)}$ and $x_0^{(2)} \to x_1^{(2)}$, and connect their intermediate states $x_t^{(1)}$ and $x_t^{(2)}$ at each timestep $t$ using dashed lines to illustrate their interactions. Under FM, the velocity remains constant, resulting in evenly spaced dashed lines. In contrast, under TFM, the trajectory accelerates after the intersection point, leading to non-uniform spacing.

We then add a third flow $x_0^{(3)} \to x_1^{(3)}$ at different locations (shown in the left and right subfigures of Figure 2) and again draw the interactions between the first two flows. We observe that introducing a third frame alters the TFM trajectories: it directly affects the flow $x_0^{(2)} \to x_1^{(2)}$ and indirectly influences $x_0^{(1)} \to x_1^{(1)}$. When the third frame is placed on the left, $x_0^{(2)} \to x_1^{(2)}$ begins with a higher velocity; when placed on the right, its initial velocity slows, which all rigorously follow Eq (6) mathematically. In contrast, FM trajectories remain unchanged regardless of where the third frame is placed, highlighting the absence of explicit temporal constraints. Further discussions on the connection between FM and TFM are provided in Appendix D.2.

In addition, despite these additional temporal interactions, TFM remains numerically stable throughout training. In our formulation, the matrix $\bar{M}$ (Eq (11)) is tridiagonal, diagonally dominant, and has bounded entries ($|\bar{M}_{i,j}| \leq \rho + 2 = 6$). These properties ensure that the operator norms of $\bar{M}$

and $\bar{M}^T$ remain bounded, which directly constrains the gradient of the magnitude term $\nabla_u J = 2\bar{M}^T(\bar{M}\bar{u} - C)$ (Eq (12)) and prevents uncontrolled gradient explosion. Meanwhile, diagonal dominance guarantees that the eigenvalues of $\bar{M}^T\bar{M}$ stay bounded away from zero, mitigating gradient vanishing across different directions. Combined with the tridiagonal structure, which avoids long-range amplification or attenuation effects, these properties ensure that gradients remain well-conditioned during optimization.

# 4. Theoretical Analysis

In this section, we theoretically investigate why our method is effective for video data by analyzing its temporal perception ability. We first make some assumptions. Let all frames in a video clip, $\{x_1^{(1)}, x_1^{(2)}, \cdots, x_1^{(n)}\}$, be sampled from the same (unknown) distribution $D$, i.e., $x_1^{(i)} \sim D$ for $1 \le i \le n$, consistent with standard FM. We further impose the following three assumptions:

**Assumption 4.1.** The first-order differences $x_1^{(i)} - x_1^{(i-1)}$, for $2 \le i \le n$ follow the same unknown distribution. Consequently, $d(x_1^{(i)}, x_1^{(i-1)})$, for $2 \le i \le n$ all follow the same distribution, denoted $D^{(1)}$.

**Assumption 4.2.** The second-order differences $x_1^{(i)} - x_1^{(i-2)}$, for $3 \le i \le n$ follow the same unknown distribution. Consequently, $d(x_1^{(i)}, x_1^{(i-2)})$, for $3 \le i \le n$ all follow the same distribution, denoted $D^{(2)}$.

**Assumption 4.3.** The two distributions introduced in Assumption 4.1 and 4.2 are distinct, i.e., $D^{(1)} \neq D^{(2)}$.

Assumption 4.1 states that the first-order temporal differences, corresponding to changes between consecutive frames, share a common unknown distribution $D^{(1)}$, which may be highly related to motion information, since motion can be interpreted as the temporal evolution of natural images. Assumption 4.2 extends this to second-order differences, assumed to follow $D^{(2)}$. Finally, Assumption 4.3 reflects the natural expectation that $D^{(1)}$ and $D^{(2)}$ are distinct, as differences between consecutive frames generally differ from those separated by larger temporal gaps. We note that these assumptions, which generally follow those in standard FM, are applicable to natural videos.

We then hypothesize that the model is trained to fit different distributions based on their training objectives. For FM, the training objective is simple, i.e., $\phi_{\theta^{\text{FM}}}^{\text{Predict}}(\psi_t(x|x_1^{(i)})) = x_1^{(i)} - x_0^{(i)}$, therefore, the prediction made by model trained under FM is expected to follow the distribution:

$$D_{\text{predict}}^{\text{FM}} \triangleq \mathcal{L}[\phi_{\theta^{\text{FM}}}^{\text{Predict}}(\psi_t(x|x_1^{(i)}))] \overset{d}{=} D * \mathcal{N}(0,1), \quad (15)$$

where $\mathcal{L}[\cdot]$ represents the distribution (law) of a random variable. While for TFM, the training objective for $\bar{M}\bar{u}_t(x) =$

$C$ (Eq (11)) can be simplified to ($1 < i < n$):

$$\phi_{\theta^{\text{TFM}}, i}^{\text{Predict}}(\psi_t(x|x_1))$$
$$\triangleq f_i\big(\bar{u}(\psi_t(x|x_1^{(i-1)})), \bar{u}(\psi_t(x|x_1^{(i)})), \bar{u}(\psi_t(x|x_1^{(i+1)}))\big)$$
$$= C_i \simeq -\rho d(x_0^{(i)}, x_1^{(i)}) + d(x_1^{(i)}, x_1^{(i+1)}) + d(x_1^{(i-1)}, x_1^{(i)})$$
$$\triangleq g_i(x) + d(x_1^{(i-1)}, x_1^{(i)}),$$
$$(16)$$

where $f_i(\cdot, \cdot, \cdot)$ is a linear combination of the three adjacent frame velocities, $g_i(x) \triangleq -\rho d(x_0^{(i)}, x_1^{(i)}) + d(x_1^{(i)}, x_1^{(i+1)})$, and we exclude the constant terms w.r.t the standard gaussian distribution that are independent of the rest in $C_i$.

Consequently, the prediction made by model trained under TFM is expected to be the following distribution:

$$D_{\text{predict}}^{\text{TFM}} \triangleq \mathcal{L}[\phi_{\theta^{\text{TFM}}, i}^{\text{Predict}}(\psi_t(x|x_1))] \overset{d}{=} H_{Y, D^{(1)}}, \quad (17)$$

$$H_{Y, D^{(1)}}(z) = \int h_{g_i(x), d(x_1^{(i-1)}, x_1^{(i)})}(a, z-a) da, \quad (18)$$

where $h_{g_i(x), d(x_1^{(i-1)}, x_1^{(i)})}(\cdot, \cdot)$ is the joint distribution of $g_i(x)$ and $d(x_1^{(i-1)}, x_1^{(i)})$, with marginal distributions $Y$ and $D^{(1)}$, respectively.

Based on the above assumptions, we can compare and measure a model's sensitivity to temporal coherence under different training objectives. In particular, perturbing the temporal structure of the data induces a training distribution $D_{\text{train}}$ that deviates from the model's fitted distribution $D_{\text{predict}}$. A model that effectively leverages temporal information should detect such deviations.

Without loss of generality, we consider a simple perturbation in which the positions of two consecutive frames are swapped, i.e., exchanging $x_1^{(i)}$ and $x_1^{(i-1)}$ for a selected $1 < i < n$, which is generalizable, as any permutation can be decomposed into a sequence of such pairwise swaps. This swap introduces temporal incoherence and should be detectable by a well-trained model capable of capturing motion-related dynamics. To measure such effect, we introduce KL divergence $\mathcal{G}_{\text{KL}} \triangleq \mathcal{D}_{\text{KL}}(D_{\text{predict}} || D_{\text{train}})$ and present the following Theorem.

**Theorem 4.4.** *For FM and TFM,*

$$\mathcal{G}_{\text{KL}}^{\text{TFM}} > 0 = \mathcal{G}_{\text{KL}}^{\text{FM}}, \quad (19)$$

*and moreover,*

$$\mathcal{G}_{\text{KL}}^{\text{TFM}} \ge \frac{1}{2} \sup_{t \in \mathbb{R}} \left| \mathbb{E}_{((x,y),(k,z)) \sim \mathcal{H}}[e^{itx}e^{ity} - e^{itk}e^{itz}] \right|^2, \quad (20)$$

*where $\mathcal{H}$ represents the joint distribution of $H_{Y, D^{(1)}}$ and $H_{Y, D^{(2)}}$.*

The lower bound can be easily estimated through numerical simulation. We perform a bootstrap resampling over $\mathcal{O}(10k)$

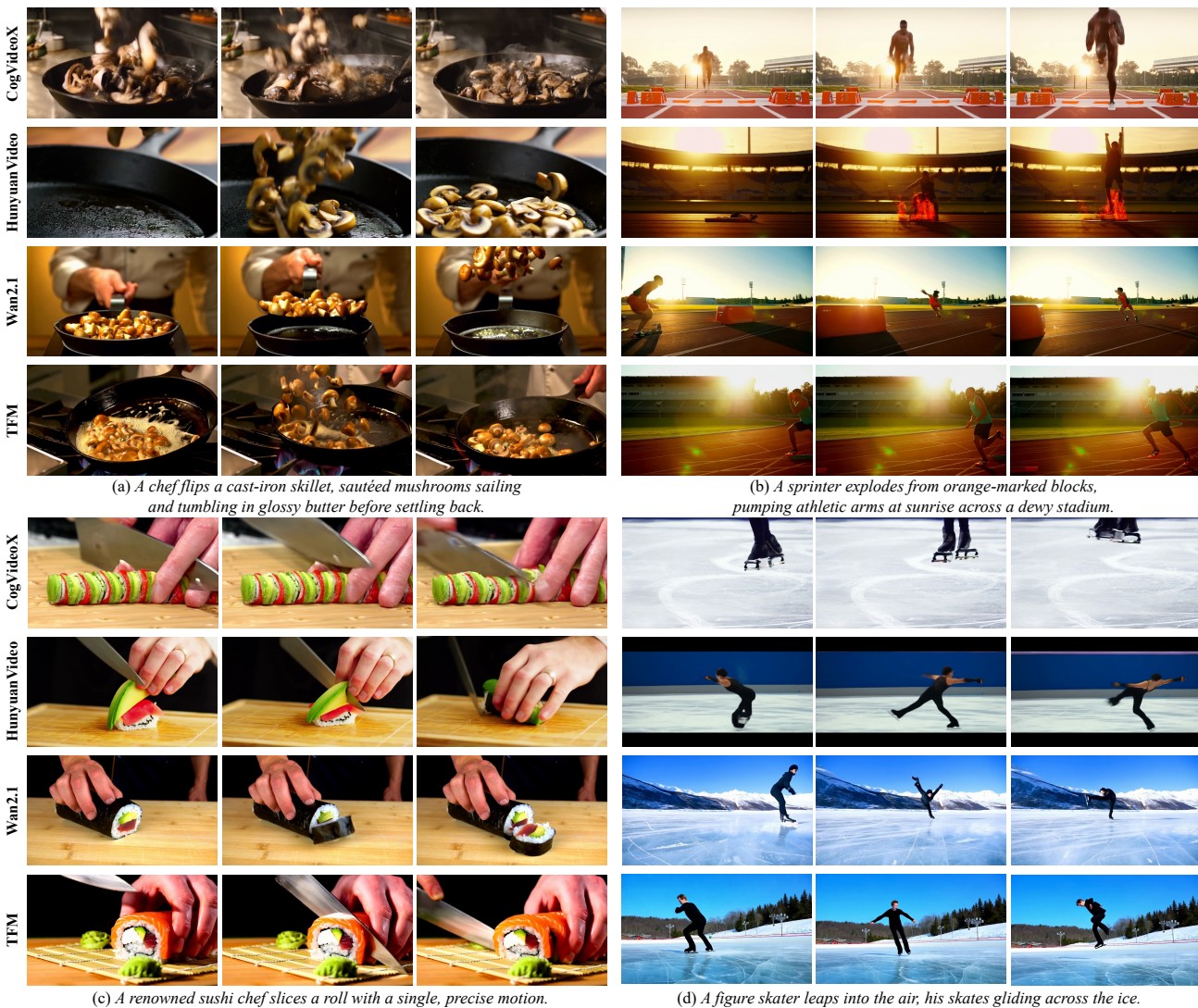

(a) *A chef flips a cast-iron skillet, sautéed mushrooms sailing and tumbling in glossy butter before settling back.*

(b) *A sprinter explodes from orange-marked blocks, pumping athletic arms at sunrise across a dewy stadium.*

(c) *A renowned sushi chef slices a roll with a single, precise motion.*

(d) *A figure skater leaps into the air, his skates gliding across the ice.*

*Figure 3.* Qualitative comparison between TFM and the baselines. We select several intermediate frames for the convenience of presentation. We provide more generated results in Appendix G, and the corresponding video samples in supplementary materials.

simulated data and derive a mean lower bound of 0.0868, with 95% confidence interval [0.0739,0.1095].

## 5. Experiments

We conduct extensive qualitative and quantitative experiments against state-of-the-art open-source video models to demonstrate the effectiveness of our method in producing temporally coherent motion. We perform ablation studies to validate the contribution of each proposed component. We then discuss the time complexity of our method and its temporal perception ability. Additional results are provided in Appendix E and Appendix G.

**Implementation Details** We adopt the pre-trained Wan2.1-T2V-14B, which is optimized using standard FM,

as our base generative model and further fine-tune it using TFM on approximately $\mathcal{O}(40k)$ data from the public ShareGPT4Video dataset (Chen et al., 2024), which accounts for only 0.01% of the original pre-training data. The model is configured to generate 81-frame videos at a fixed resolution of $832 \times 480$ with fps = 15. Further implementation details are provided in Appendix B.

**Evaluation Details** We compare our method with the following baselines: (1) CogVideoX1.5-5B (Yang et al., 2024), (2) HunyuanVideo-13B (Wu et al., 2025), and (3) Wan2.1-T2V-14B (Wan et al., 2025), all of which are recent and competitive video generation models. We evaluate performance on two benchmarks: (1) VideoJam-Bench (Chefer et al., 2025), which emphasizes challenging motion, and (2) Movie-Gen (Polyak et al., 2024), a general-purpose video

*Table 1.* Quantitative comparison between TFM and the baselines on VideoJam-Bench. User study shows percentage of votes favoring TFM. See breakdown of VBench metrics in Appendix G.

| Method | Auto. Metric | | User Study | | |
| --- | --- | --- | --- | --- | --- |
| | Appearance | Motion | Semantic | Quality | Motion |
| CogVideoX1.5 | 0.750 | 0.894 | 87.8% | 85.4% | 82.9% |
| HunyuanVideo | 0.752 | 0.898 | 65.6% | 56.2% | 68.8% |
| Wan2.1 | 0.780 | 0.888 | 82.8% | 85.7% | 74.3% |
| **TFM** | **0.783** | **0.961** | - | - | - |

*Table 2.* Quantitative comparison between TFM and the baselines on Movie-Gen. User study shows percentage of votes favoring TFM. See breakdown of VBench metrics in Appendix G.

| Method | Auto. Metric | | User Study | | |
| --- | --- | --- | --- | --- | --- |
| | Appearance | Motion | Semantic | Quality | Motion |
| CogVideoX1.5 | 0.783 | 0.801 | 90.9% | 86.3% | 88.6% |
| HunyuanVideo | 0.775 | 0.806 | 76.9% | 53.8% | 88.5% |
| Wan2.1 | 0.791 | 0.805 | 58.3% | 62.5% | 64.6% |
| **TFM** | **0.795** | **0.858** | - | - | - |

generation benchmark. We adopt two types of evaluation metrics: (1) automatic metrics supported by VBench (Huang et al., 2024), and (2) human evaluations following the two-alternative forced-choice (2AFC) protocol, where raters compare two videos (one generated by TFM and the other by a baseline) and select the preferred one across multiple aspects, following prior work (Blattmann et al., 2023a). Further evaluation details are provided in Appendix C.

### 5.1. Main Results

**Qualitative Experiment** In Figure 3, we compare TFM with baseline methods across a variety of motion types. These include intricate motions requiring physical understanding (the left column), such as flipping a cast-iron skillet, where mushrooms are expected to be propelled into the air and fall back smoothly, and slicing a roll, where the knife should cut cleanly through the roll to separate a slice. We also evaluate complex human motions (the right column), such as a sprinter running or a figure skater dancing, which pose significant challenges for video generative models.

From the results, we observe that baseline methods generally fail to respect basic physical laws. For example, in Figure 3(a), Wan2.1 generates mushrooms flying upward while the skillet itself remains static. Similarly, in Figure 3(c), CogVideoX and HunyuanVideo produce physically implausible motions, where the knife appears to pass through a finger or the roll as if the objects were intangible. For complex human motions (Figure 3(c) and (d)), baseline models frequently generate deformed body structures. A common failure mode is that different body parts (*e.g.*, legs or arms) move independently of the torso, resulting in unnatural poses where the upper body and lower body face

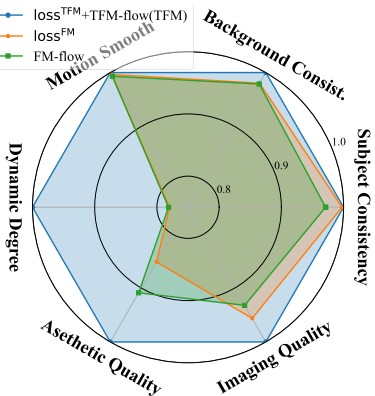

*Figure 4.* Ablation study. We ablate the choice of training loss (*i.e.*, residual-type $loss^{TFM}$ vs. solution-type $loss^{FM}$) and the flow formulation used during fine-tuning (*i.e.*, standard FM vs. our proposed TFM). Evaluation is conducted on Movie-Gen using VBench metrics. All results are reported as ratios relative to the full TFM model; exact statistics are provided in Appendix G.

different directions. In contrast, models trained with TFM generate more coherent and physically plausible motions across a wide range of scenarios. These results demonstrate TFM's ability to capture motion priors from training data effectively, leading to improved temporal consistency.

**Quantitative Experiment** In Tables 1 and 2, we quantitatively compare TFM with baseline methods. For fairness, all models are evaluated using a fixed random seed and a single run on both benchmarks. Overall, TFM consistently outperforms the baselines across the two benchmarks. On general metrics (*Appearance*, *Semantic*, and *Quality*), TFM achieves performance that is competitive with or superior to its base model and other state-of-the-art methods. More importantly, TFM achieves substantial improvements on the motion metric, with a 74.3% higher preference rate on VideoJam-Bench and a 64.6% higher preference rate on the Movie-Gen benchmark compared to its base model. It also beats other baseline methods in terms of motion coherence. These quantitative improvements corroborate the qualitative observations, demonstrating that TFM more effectively models motion dynamics and enhances temporal control in generated videos. Moreover, they indicate that improving motion modeling does not come at the expense of other objectives, such as visual quality or semantic coherence.

### 5.2. Ablation Study

In this section, we conduct an ablation study to assess the contribution of each component in our method. Specifically, we first fine-tune the base model using standard FM (denoted as *FM-flow*) under the same training settings and on the same data as TFM, thereby isolating the effect of the proposed flow formulation. Next, we ablate the choice of loss function by training TFM with the solution-type $loss^{FM}$, while keeping all other settings unchanged. The results are

*Table 3.* Comparison of time costs between FM and TFM. We consider two metrics: (1) *Net Time Cost*, which measures the time spent on flow calculation, velocity prediction, and loss computation, corresponding to lines 17–18 in Algorithm 1 for FM, and lines 28–29 in Algorithm 2 for TFM; and (2) *Overall Time Cost*, which records the total time for a single optimization step for each model. We report the mean time cost, with standard errors indicated in parentheses.

| Model | Net Time Cost (FM) | Net Time Cost (TFM) | Overall Time Cost (FM) | Overall Time Cost (TFM) |
|---|---|---|---|---|
| Wan2.1-T2V-1.3B | 1.65s($\pm$0.33s) | 2.85s($\pm$0.37s) 
 +72.7% | 18.71s($\pm$0.9s) | 19.66s($\pm$0.47s) 
 +5.1% |
| Wan2.1-T2V-14B | 10.67s($\pm$0.85s) | 12.98s($\pm$0.39s) 
 +21.6% | 60.51s($\pm$0.78s) | 61.92s($\pm$0.98s) 
 +2.3% |

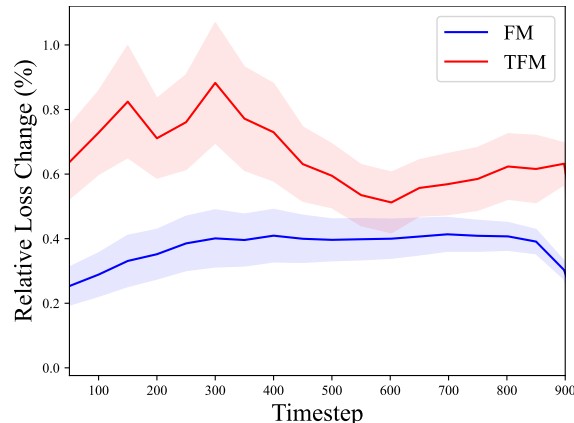

*Figure 5.* Quantitative comparison of the temporal perception abilities of FM and TFM. The solid line shows the mean loss-increase ratio after randomly permuting video frames, and the shaded region indicates the corresponding standard deviation.

summarized in Figure 4, where the blue curve corresponds to the full TFM model. From the results, we observe that the full TFM consistently outperforms both variants, particularly on *Dynamic Degree*, which directly reflects the magnitude of motion. In contrast, the two ablated variants exhibit relatively high *Motion Smooth*, likely due to their tendency to generate more static or under-dynamic motions. Importantly, TFM also maintains strong performance on general appearance-related metrics. Overall, these findings demonstrate that both the temporal flow formulation and the residual-type loss design are essential for achieving realistic and dynamic motion without compromising visual quality.

### 5.3. Time Complexity Analysis

In this section, we demonstrate that the computational overhead introduced by our method is marginal relative to the overall optimization cost. The results are presented in Table 3, where we compare time costs between FM and TFM across models of different scales using $\mathcal{O}(1k)$ data. Although TFM exhibits a seemingly large increase in *Net Time Cost* for Wan2.1-T2V-1.3B, the overhead is minimal when considering the total time per optimization step, amounting to only 5.1%. For the larger Wan2.1-T2V-14B model, this relative increase reduces further to 2.3%. These results con-

firm that TFM introduces only a negligible time overhead, thanks to the efficient optimization algorithm and necessary simplifications in the formulas.

### 5.4. Temporal Perception Ability

In Section 4, we establish the theoretical basis for why TFM better suits video data than standard FM, showing that explicit temporal constraints enhance a model's ability to perceive temporal structure. Here, we further illustrate this claim through an intuitive quantitative experiment. Specifically, for each video, we apply noise to a chosen timestep $t \in \{50, 100, \cdots, 900\}$ out of 1000 total steps, and then randomly permute the video latents along the temporal dimension. For every timestep, we simulate $\mathcal{O}(1k)$ samples. A model with strong temporal perception should exhibit a larger loss increase after permutation, whereas a model with weak temporal understanding will struggle to distinguish a correctly ordered video from a permuted one.

We therefore compute the mean and standard deviation of the relative loss change before and after permutation, and present the results in Figure 5. For *FM* (the blue line), we use the pre-trained Wan2.1-T2V-1.3B model, while for *TFM* (the red line), we use the same model fine-tuned with TFM. As shown, the FM-trained model displays only limited temporal sensitivity, while TFM substantially boosts the model's ability to detect temporal disruptions. This experiment further validates both the effectiveness of TFM and the importance of explicitly modeling temporal constraints.

## 6. Conclusion

In this paper, we introduce Temporal-aware Flow Matching (TFM), a method that enhances motion generation by explicitly modeling temporal structure in video data. Unlike prior approaches that rely on implicit attention mechanisms, TFM provides a principled flow formulation for capturing motion dynamics. We theoretically show that TFM improves temporal perception ability and empirically demonstrate consistent performance gains across benchmarks. As a model-agnostic approach with no additional inference cost, TFM enables more effective motion modeling for video generation.

## Acknowledgements

This work is supported by the National Key Research and Development Program of China under Grant No.2022ZD0115903, National Natural Science Foundation of China No.62222209, Beijing National Research Center for Information Science and Technology under Grant No.BNR2026TD03005.

## Impact Statement

This work aims to improve motion modeling in video generation by enhancing temporal coherence through Temporal-aware Flow Matching, enabling generative models to produce more consistent and realistic motion. As with other advances in generative video technologies, the proposed method could potentially be misused to create misleading visual content; however, it does not introduce new risks beyond those inherent to existing video generation models. We believe that continued efforts in responsible deployment, bias detection, and misuse mitigation are essential to ensure the safe and fair use of generative video systems, including those based on our approach.

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

## A. Theorem Proofs

**Proposition** 3.1: *Eq (6) ($\rho \geq 4$) characterizes flows that reach $x_1^{(i)}$ at $t = 1$, for all $1 \leq i \leq n$.*

*Proof.* Set $t = 1$ in Eq (6), and we get

$$\rho \cdot d(\psi_1(x|x_1^{(i)}), x_1^{(i)}) + \mathbb{I}_{i>1} d(\psi_1(x|x_1^{(i-1)}), \psi_1(x|x_1^{(i)})) + \mathbb{I}_{i<n} d(\psi_1(x|x_1^{(i)}), \psi_1(x|x_1^{(i+1)}))$$
$$= \mathbb{I}_{i>1} d(x_1^{(i-1)}, x_1^{(i)}) + \mathbb{I}_{i<n} d(x_1^{(i)}, x_1^{(i+1)}). \tag{21}$$

Summation over $1 \leq i \leq n$, we get

$$2 \sum_{i=1}^{n-1} d(x_1^{(i)}, x_1^{(i+1)}) = \sum_{i=1}^{n} \mathbb{I}_{i>1} d(x_1^{(i-1)}, x_1^{(i)}) + \mathbb{I}_{i<n} d(x_1^{(i)}, x_1^{(i+1)})$$

$$= \sum_{i=1}^{n} \rho \cdot d(\psi_1(x|x_1^{(i)}), x_1^{(i)}) + \mathbb{I}_{i>1} d(\psi_1(x|x_1^{(i-1)}), \psi_1(x|x_1^{(i)})) + \mathbb{I}_{i<n} d(\psi_1(x|x_1^{(i)}), \psi_1(x|x_1^{(i+1)}))$$

$$= \sum_{i=1}^{n} \rho \cdot d(\psi_1(x|x_1^{(i)}), x_1^{(i)}) + 2 \sum_{i=1}^{n-1} d(\psi_1(x|x_1^{(i)}), \psi_1(x|x_1^{(i+1)})) \tag{22}$$

$$\geq 2 \sum_{i=1}^{n-1} d(\psi_1(x|x_1^{(i)}), x_1^{(i)}) + d(\psi_1(x|x_1^{(i)}), \psi_1(x|x_1^{(i+1)})) + d(\psi_1(x|x_1^{(i+1)}), x_1^{(i+1)})$$

$$\geq 2 \sum_{i=1}^{n-1} d(x_1^{(i)}, x_1^{(i+1)}),$$

where the last inequality is due to the *Triangle Inequality* constraint. Note that the derivation relies on the symmetry of the metric function, *i.e.*, $d(z_1, z_2) = d(z_2, z_1)$. This property is naturally satisfied by the Euclidean distance used in TFM formulation. For non-symmetric metric functions, one can instead consider the symmetrized form $d'(z_1, z_2) = \frac{1}{2}\big[d(z_1, z_2) + d(z_2, z_1)\big]$ to maintain symmetry.

Then Eq (22) becomes an equality if and only if $\psi_1(x|x_1^{(i)}) = x_1^{(i)}$, for $1 \leq i \leq n$, which completes the proof. □

**Theorem** 3.2: $u_t(x) = \big[u(\psi_t(x|x_1^{(1)})) \quad \cdots \quad u(\psi_t(x|x_1^{(n)}))\big]^T$ *generates the flow defined in Eq (6), if and only if it satisfies the following ODE system $Mu_t(x) = C$, where*

$$M = \begin{bmatrix} \rho A_1^T \\ +B_{1,2}^T & B_{2,1}^T & \cdots \\ & \ddots \\ \cdots & B_{i-1,i}^T & \rho A_i^T & B_{i+1,i}^T & \cdots \\ & +B_{i,i-1}^T \\ & +B_{i,i+1}^T \\ & \ddots \\ & \cdots & B_{n-1,n}^T & \rho A_n^T \\ & & & +B_{n,n-1}^T \end{bmatrix}, \tag{23}$$

*and $C = [C_1, \cdots, C_n]^T$.*

*Proof.* Recall that (Eq (7))

$$\begin{cases} A_i = \nabla_x d(\psi_t(x|x_1^{(i)}), x_1^{(i)}), \ 1 \leq i \leq n, \ \text{(intra-frame } i) \\ B_{i,j} = \nabla_x d(\psi_t(x|x_1^{(i)}), \psi_t(x|x_1^{(j)})) \\ B_{j,i} = \nabla_y d(\psi_t(x|x_1^{(i)}), \psi_t(x|x_1^{(j)})) \end{cases} \begin{matrix} 1 \leq i < j \leq n, \\ \text{(inter-frame } i \leftrightarrow j) \end{matrix} \\ C_i = \frac{d}{dt} \tau(t) \begin{bmatrix} \rho d(x_0^{(i)}, x_1^{(i)}) + \mathbb{I}_{i>1}\big(d(x_0^{(i-1)}, x_0^{(i)}) - d(x_1^{(i-1)}, x_1^{(i)})\big) \\ + \mathbb{I}_{i<n}\big(d(x_0^{(i)}, x_0^{(i+1)}) - d(x_1^{(i)}, x_1^{(i+1)})\big), 1 \leq i \leq n \end{bmatrix}, \tag{24}$$

First, we prove that the solution $u_t(x)$ to Eq (23) generates the flow defined by Eq (6). Differentiate the left side of Eq (6) for any $1 \leq i \leq n$ with respect to time gives:

$$
\begin{aligned}
&\frac{d}{dt}[\rho \cdot d(\psi_t(x|x_1^{(i)}), x_1^{(i)}) + \mathbb{I}_{i>1}d(\psi_t(x|x_1^{(i-1)}), \psi_t(x|x_1^{(i)})) + \mathbb{I}_{i<n}d(\psi_t(x|x_1^{(i)}), \psi_t(x|x_1^{(i+1)}))] \\
&=[\rho A_i + \mathbb{I}_{i>1}B_{i,i-1} + \mathbb{I}_{i<n}B_{i,i+1}]^T u(\psi_t(x|x_1^{(i)})) + \mathbb{I}_{i>1}B_{i-1,i}^T u(\psi_t(x|x_1^{(i-1)})) + \mathbb{I}_{i<n}B_{i+1,i}^T u(\psi_t(x|x_1^{(i+1)})) \\
&=C_i.
\end{aligned}
\tag{25}
$$

Thus the function $\mathcal{K}_i(t) = \rho \cdot d(\psi_t(x|x_1^{(i)}), x_1^{(i)}) + \mathbb{I}_{i>1}d(\psi_t(x|x_1^{(i-1)}), \psi_t(x|x_1^{(i)})) + \mathbb{I}_{i<n}d(\psi_t(x|x_1^{(i)}), \psi_t(x|x_1^{(i+1)}))$ satisfies the ODE:

$$
\frac{d}{dt}\mathcal{K}_i(t) = C_i,
\tag{26}
$$

with initial condtion

$$
\mathcal{K}_i(0) = \rho \cdot d(x_0^{(i)}, x_1^{(i)}) + \mathbb{I}_{i>1}d(x_0^{(i-1)}, x_0^{(i)}) + \mathbb{I}_{i<n}d(x_0^{(i)}, x_0^{(i+1)}),
\tag{27}
$$

where by default we start sampling from noise $x_0 = \{x_0^{(1)}, x_0^{(2)}, \cdots, x_0^{(n)}\}$. The general solution to the ODE defined by Eq (26) has the form

$$
\mathcal{K}_i(t) = \int C_i dt + C_i',
\tag{28}
$$

where $C_i'$ is a constant constrained by initial conditions. Substitute Eq (27) into Eq (28), we can verify that

$$
\begin{aligned}
\mathcal{K}_i(t) =& \rho \cdot \tau(t)d(x_0^{(i)}, x_1^{(i)}) + \\
& \mathbb{I}_{i>1}[\tau(t)d(x_0^{(i-1)}, x_0^{(i)}) + (1 - \tau(t))d(x_1^{(i-1)}, x_1^{(i)})] + \\
& \mathbb{I}_{i<n}[\tau(t)d(x_0^{(i)}, x_0^{(i+1)}) + (1 - \tau(t))d(x_1^{(i)}, x_1^{(i+1)})],
\end{aligned}
\tag{29}
$$

which is exactly the right side of Eq (6).

Conversely, suppose the flow defined by Eq (6) is $x_t = \{x_t^{(1)}, x_t^{(2)}, \cdots, x_t^{(n)}\}$, where $x_t^{(i)} = \psi_t(x|x_1^{(i)})$, for any $1 \leq i \leq n$. Differentiating both side of Eq (6) we get:

$$
\begin{aligned}
&\frac{d}{dt}[\rho \cdot d(\psi_t(x|x_1^{(i)}), x_1^{(i)}) + \mathbb{I}_{i>1}d(\psi_t(x|x_1^{(i-1)}), \psi_t(x|x_1^{(i)})) + \mathbb{I}_{i<n}d(\psi_t(x|x_1^{(i)}), \psi_t(x|x_1^{(i+1)}))] \\
&=[\rho A_i + \mathbb{I}_{i>1}B_{i,i-1} + \mathbb{I}_{i<n}B_{i,i+1}]^T \dot{x}_t^{(i)} + \mathbb{I}_{i>1}B_{i-1,i}^T \dot{x}_t^{(i-1)} + \mathbb{I}_{i<n}B_{i+1,i}^T \dot{x}_t^{(i+1)},
\end{aligned}
\tag{30}
$$

and

$$
\begin{aligned}
&\frac{d}{dt}\{\rho \cdot \tau(t)d(x_0^{(i)}, x_1^{(i)}) + \mathbb{I}_{i>1}[\tau(t)d(x_0^{(i-1)}, x_0^{(i)}) + (1 - \tau(t))d(x_1^{(i-1)}, x_1^{(i)})] + \\
& \mathbb{I}_{i<n}[\tau(t)d(x_0^{(i)}, x_0^{(i+1)}) + (1 - \tau(t))d(x_1^{(i)}, x_1^{(i+1)})]\} \\
&=C_i.
\end{aligned}
\tag{31}
$$

Combining the two equations above, we get

$$
[\rho A_i + \mathbb{I}_{i>1}B_{i,i-1} + \mathbb{I}_{i<n}B_{i,i+1}]^T \dot{x}_t^{(i)} + \mathbb{I}_{i>1}B_{i-1,i}^T \dot{x}_t^{(i-1)} + \mathbb{I}_{i<n}B_{i+1,i}^T \dot{x}_t^{(i+1)} = C_i,
\tag{32}
$$

which is exactly $M\dot{x}_t = C$. $\qquad\square$

In practice, we set $\tau(t) = 1 - t$, use the Euclidean distance for $d(\cdot, \cdot)$, and constrain each direction $u(\psi_t(x \mid x_1^{(i)}), x_1^{(i)})$ to align with $x_1^{(i)} - x_0^{(i)}$. Under these choices, the system reduces to an ODE of the form $\bar{M}\bar{u}_t(x) = C$ (Eq (11)), where $\bar{M} \in \mathbb{R}^{n \times n}$ becomes a tridiagonal matrix, with

$$
\begin{cases}
\bar{M}_{i,i-1} = \hat{v}^{(i-1,i)T}\hat{v}^{(i-1)}, i > 1, \\
\bar{M}_{i,i} = \mathbb{I}_{i>1}\hat{v}^{(i,i-1)T}\hat{v}^{(i)} + \mathbb{I}_{i<n}\hat{v}^{(i,i+1)T}\hat{v}^{(i)} - \rho, 1 \leq i \leq n, \\
\bar{M}_{i,i+1} = \hat{v}^{(i+1,i)T}\hat{v}^{(i+1)}, i < n.
\end{cases}
\tag{33}
$$

For $\rho = 4.0$, we have

$$
\begin{aligned}
|\bar{M}_{i,i}| &= |\mathbb{I}_{i>1}\hat{v}^{(i,i-1)T}\hat{v}^{(i)} + \mathbb{I}_{i<n}\hat{v}^{(i,i+1)T}\hat{v}^{(i)} - \rho| \\
&\geq \rho - 2 = 2 \\
&\geq |\mathbb{I}_{i>1}\hat{v}^{(i-1,i)T}\hat{v}^{(i-1)}| + |\mathbb{I}_{i<n}\hat{v}^{(i+1,i)T}\hat{v}^{(i+1)}| \\
&= |\mathbb{I}_{i>1}\bar{M}_{i,i-1}| + |\mathbb{I}_{i<n}\bar{M}_{i,i+1}|.
\end{aligned}
\tag{34}
$$

For the inequality to become equality (for $1 < i < n$), $\hat{v}^{(i-1,i)}$ must be aligned with $\hat{v}^{(i-1)}$, and $\hat{v}^{(i,i-1)}$, that is the opposite direction of $\hat{v}^{(i-1,i)}$, must be aligned with $\hat{v}^{(i)}$. This forces $\hat{v}^{(i-1)}$ and $\hat{v}^{(i)}$ to be collinear. By the same reasoning, $\hat{v}^{(i)}$ and $\hat{v}^{(i+1)}$ must also be collinear. As a result, the samples $x_1^{(i-1)}, x_1^{(i)}, x_1^{(i+1)}$ and the independently drawn noises $x_0^{(i-1)}, x_0^{(i)}, x_0^{(i+1)}$ would all lie on the same line, which is an event with probability zero and thus negligible.

Therefore, for $\rho = 4.0$, we conclude that $\bar{M}$ is diagonally dominant. This property is crucial: it guarantees that the system $\bar{M}\bar{u}_t(x) = C$ admits a unique solution and ensures that the Thomas algorithm (Thomas, 1949; Lee, 2011) remains numerically stable.

**Proposition** 3.4: $\mathrm{loss}^{\mathrm{FM}}$ *is **not** a consistent objective.*

*Proof.* For some $\theta_1, \theta_2 \in \mathcal{M}$ and $\bar{u}_{\theta_1}, \bar{u}_{\theta_2}$ the predictions, denote $\epsilon^{(1)} = \bar{u}_{\theta_1} - \bar{u}^*$ and $\epsilon^{(2)} = \bar{u}_{\theta_2} - \bar{u}^*$, where $\bar{u}^*$ is the unique solution to $\bar{M}\bar{u} = C$.

To prove Proposition 3.4, according to Definition 3.3, we only need to show that $\mathrm{loss}^{\mathrm{FM}}(\bar{u}_{\theta_1}) = \mathrm{loss}^{\mathrm{FM}}(\bar{u}_{\theta_2})$, while $\Lambda(\bar{u}_{\theta_1}) > \Lambda(\bar{u}_{\theta_1})$. This is equivalent to the following statement:

$$
||\epsilon^{(1)}||_2^2 = ||\epsilon^{(2)}||_2^2, \text{ while } ||\bar{M}\epsilon^{(1)}||_2^2 > ||\bar{M}\epsilon^{(2)}||_2^2.
\tag{35}
$$

Since $\bar{M}$ is a diagonally dominant tridiagonal matrix, and $\bar{M} \neq cI$ for any $c \in \mathbb{R}$, $\bar{M}^T\bar{M}$ becomes diagonalizable and has at least two different eigenvalues, denoted as $\lambda_{\max} > \lambda_{\min}$, with corresponding unit eigenvectors $e_{\max}$ and $e_{\min}$. Therefore, set $\epsilon^{(1)} = e_{\max}$, and $\epsilon^{(2)} = e_{\min}$, and we get

$$
||e_{\max}||_2^2 = ||e_{\min}||_2^2, \text{ while } ||\bar{M}e_{\max}||_2^2 = \lambda_{\max} > \lambda_{\min} = ||\bar{M}e_{\min}||_2^2,
\tag{36}
$$

which completes the proof. $\qquad\square$

**Example** For Proposition 3.4, we give an intuitive and concrete example to better illustrate it. Consider a simple system of two linear equations with two variables:

$$
\begin{cases} x + y = 5 \\ 2x + y = 8 \end{cases},
\tag{37}
$$

which gives $x = 3$ and $y = 2$. Consider the following two cases:

1. A prediction $(x_1, y_1)$ of $x_1 = 2$ and $y_1 = 1$, which gives $x_1 + y_1 = 3$ and $2x_1 + y_1 = 5$. In this case, the residual-type $\mathrm{loss}_1^{\mathrm{TFM}}$ gives $\frac{1}{2}(4+9) = 6.5$, while solution-type $\mathrm{loss}_1^{\mathrm{FM}}$ gives $\frac{1}{2}(1+1) = 1$.

2. A prediction $(x_2, y_2)$ of $x_2 = 4$ and $y_2 = 1$, which gives $x_2 + y_2 = 5$ and $2x_2 + y_2 = 9$. In this case, the residual-type $\mathrm{loss}_2^{\mathrm{TFM}}$ gives $\frac{1}{2}(0+1) = 0.5$, while solution-type $\mathrm{loss}_2^{\mathrm{FM}}$ gives $\frac{1}{2}(1+1) = 1$.

We see that obviously $(x_2, y_2)$ better fits the linear equations, while it is considered *equal* as $(x_1, y_1)$ under $\mathrm{loss}^{\mathrm{FM}}$, since $\mathrm{loss}_1^{\mathrm{FM}} = \mathrm{loss}_2^{\mathrm{FM}}$. This phenomenon highlights that $\mathrm{loss}^{\mathrm{FM}}$ will be less effective in training compared to $\mathrm{loss}^{\mathrm{TFM}}$, although they share the common global minimum.

**Theorem** 4.4: *For FM and TFM,*

$$
\mathcal{G}_{\mathrm{KL}}^{\mathrm{TFM}} > 0 = \mathcal{G}_{\mathrm{KL}}^{\mathrm{FM}},
\tag{38}
$$

*and moreover,*

$$
\mathcal{G}_{\mathrm{KL}}^{\mathrm{TFM}} \geq \frac{1}{2} \sup_{t \in \mathbb{R}} \left| \mathbb{E}_{((x,y),(k,z)) \sim \mathcal{H}} \left[ e^{itx}e^{ity} - e^{itk}e^{itz} \right] \right|^2,
\tag{39}
$$

*where $\mathcal{H}$ represents the joint distribution of $H_{Y,D^{(1)}}$ and $H_{Y,D^{(2)}}$.*

*Proof.* Recall that we perturb the temporal structure of the data by swapping $x_1^{(i)}$ and $x_1^{(i-1)}$, for a selected $2 < i < n$. For FM, its training objective changes to $\phi_{\theta^{\text{FM}},i}^{\text{Train}} = \phi_{\theta^{\text{FM}}}^{\text{Train}}(\phi_t(x|x_1^{(i-1)})) = x_1^{(i-1)} - x_0^{(i)} \stackrel{d}{=} x_1^{(i-1)} - x_0^{(i-1)}$. Therefore, the same $D_{\text{train}}^{\text{FM}} \stackrel{d}{=} D * \mathcal{N}(0,1) \stackrel{d}{=} D_{\text{predict}}^{\text{FM}}$ is obtained, thus $\mathcal{G}_{\text{KL}}^{\text{FM}} = \mathcal{D}_{KL}(D_{\text{predict}}^{\text{FM}}||D_{\text{train}}^{\text{FM}}) = 0$.

For TFM, swapping $x_1^{(i)}$ and $x_1^{(i-1)}$ will change its training objective to

$$
\begin{aligned}
\phi_{\theta^{\text{TFM}},i}^{\text{Train}}(\psi_t(x|x_1)) &= -\rho d(x_0^{(i)}, x_1^{(i-1)}) + d(x_1^{(i-1)}, x_1^{(i+1)}) + d(x_1^{(i-1)}, x_1^{(i)}) \\
&\stackrel{d}{=} -\rho d(x_0^{(i-1)}, x_1^{(i-1)}) + d(x_1^{(i-1)}, x_1^{(i+1)}) + d(x_1^{(i-1)}, x_1^{(i)}) \\
&= -\rho d(x_0^{(i-1)}, x_1^{(i-1)}) + d(x_1^{(i-1)}, x_1^{(i)}) + d(x_1^{(i-1)}, x_1^{(i+1)}) \\
&\stackrel{d}{=} -\rho d(x_0^{(i)}, x_1^{(i)}) + d(x_1^{(i)}, x_1^{(i+1)}) + d(x_1^{(i)}, x_1^{(i+2)}) \\
&= g_i(x) + d(x_1^{(i)}, x_1^{(i+2)}). \quad \text{(If } i+2 > n, \text{ we can just assume an } x_1^{(n+1)} \text{ which extends the video.)}
\end{aligned}
\tag{40}
$$

Therefore, we have

$$
D_{\text{train}}^{\text{TFM}} \triangleq \mathcal{L}[\phi_{\theta^{\text{TFM}},i}^{\text{Train}}(\psi_t(x|x_1))] \stackrel{d}{=} H_{Y,D^{(2)}}.
\tag{41}
$$

According to Assumption 4.3, we have $D_{\text{predict}}^{\text{TFM}} \stackrel{d}{=} H_{Y,D^{(1)}} \neq H_{Y,D^{(2)}} \stackrel{d}{=} D_{\text{train}}^{\text{TFM}}$, and consequently,

$$
\mathcal{G}_{\text{KL}}^{\text{TFM}} = \mathcal{D}_{KL}(D_{\text{predict}}^{\text{TFM}}||D_{\text{train}}^{\text{TFM}}) = \mathcal{D}_{KL}(H_{Y,D^{(1)}}||H_{Y,D^{(2)}}) > 0.
\tag{42}
$$

Next we will derive a lower bound of $\mathcal{G}_{\text{KL}}^{\text{TFM}}$:

$$
\begin{aligned}
\mathcal{G}_{\text{KL}}^{\text{TFM}} &= \mathcal{D}_{KL}(D_{\text{predict}}^{\text{TFM}}||D_{\text{train}}^{\text{TFM}}) \\
&\geq 2 \cdot \text{TV}^2(D_{\text{predict}}^{\text{TFM}}, D_{\text{train}}^{\text{TFM}}) \\
&\geq \frac{1}{2}|\Phi_{H_{Y,D^{(1)}}}(t) - \Phi_{H_{Y,D^{(2)}}}(t)|^2 \\
&\geq \frac{1}{2} \sup_{t \in R} |\Phi_{H_{Y,D^{(1)}}}(t) - \Phi_{H_{Y,D^{(2)}}}(t)|^2,
\end{aligned}
\tag{43}
$$

where the first inequality is the direct result of Pinsker's inequality (Csiszár & Körner, 2011), and $\text{TV}(\cdot, \cdot)$ represents the total variance distance function. $\Phi_{H_{Y,D^{(1)}}}$ and $\Phi_{H_{Y,D^{(2)}}}$ are the characteristic functions of distribution $H_{Y,D^{(1)}}$ and $H_{Y,D^{(2)}}$. The second inequality is due to the fact that for any distribution $p$ and $q$,

$$
\begin{aligned}
\text{TV}(p, q) &= \frac{1}{2} \int_{-\infty}^{\infty} |p(x) - q(x)| dx \\
&= \frac{1}{2} \int_{-\infty}^{\infty} |e^{itx}||p(x) - q(x)| dx \\
&\geq \frac{1}{2} |\int_{-\infty}^{\infty} e^{itx}(p(x) - q(x)) dx| = \frac{1}{2}|\Phi_p(t) - \Phi_q(t)|, \forall t \in \mathbb{R}.
\end{aligned}
\tag{44}
$$

If we substitute the definition of $H_{Y,D^{(1)}}$ and $H_{Y,D^{(2)}}$ into Eq (43), we get

$$
\mathcal{G}_{\text{KL}}^{\text{TFM}} \geq \frac{1}{2} \sup_{t \in \mathbb{R}} \left| \mathbb{E}_{((x,y),(k,z)) \sim \mathcal{H}}[e^{itx}e^{ity} - e^{itk}e^{itz}] \right|^2,
\tag{45}
$$

where $\mathcal{H}$ represents the joint distribution of $H_{Y,D^{(1)}}$ and $H_{Y,D^{(2)}}$. $\square$

## B. Implementation Details

In this section, we provide implementation details of TFM to facilitate reproducibility. The code will be released upon publication. While the core ideas of our method have been fully described in the main paper and can be readily implemented on most video generative models, we further present the training procedure of TFM in Algorithm 2, together with its FM counterpart in Algorithm 1 for comparison.

Both algorithms share the same MAIN FUNCTION, in which the training process is abstracted into two stages, namely GETLATENT and GETLOSS. Notably, for FM, the training pipeline is identical to that used in image generation and does not explicitly account for temporal structure. In contrast, TFM constructs flows that explicitly model temporal dynamics, thereby providing more appropriate supervision for video generative models.

---

**Algorithm 1** Training procedure of **FM** under text-to-video modeling.

---

1: **Input**: Video data: $\left\{ x_1 = \begin{bmatrix} x_1^{(1)} & \cdots & x_1^{(n)} \end{bmatrix} \right\}_N$, paired text prompt: $\{h\}_N$.
2: **Parameter**: $\theta$: parameters for the base video generative model.

3: **function** GETVELOCITY($x_0, x_1, x_t, t$)
4:     **return** $x_1 - x_0$.
5: **end function**

6: **function** GETLATENT($x_0, x_1, t$)
7:     **return** $tx_1 + (1-t)x_0$.
8: **end function**

9: **function** GETLOSS($x_0, x_1, x_t, t, h$)
10:     $v \leftarrow$ GETVELOCITY($x_0, x_1, x_t, t$).
11:     **return** $||u_\theta(x_t, h) - v||_2^2$.
12: **end function**

    BEGIN MAIN FUNCTION:

13: **repeat**
14:     Forward data $\{x_1, h\}$.
15:     Sample $x_0 \sim \mathcal{N}(0,1)$.
16:     Sample $t \sim \mathcal{U}(0,1)$.
17:     $x_t \leftarrow$ GETLATENT($x_0, x_1, t$).
18:     loss$\leftarrow$ GETLOSS($x_0, x_1, x_t, t, h$).
19:     $\theta \leftarrow \arg\min_\theta(\text{loss})$.
20: **until** Converged

---

**Algorithm 2** Training procedure of **TFM** under text-to-video modeling.

---

1: **Input**: Video data: $\left\{ x_1 = \begin{bmatrix} x_1^{(1)} & \cdots & x_1^{(n)} \end{bmatrix} \right\}_N$, paired text prompt: $\{h\}_N$.
2: **Parameter**: $\theta$: parameters for the base video generative model.

3: **function** GETVELOCITY($x_0, x_1, x_t, t$)
4:     Compute $\bar{M}$ and $C$ according to Eq (11).
5:     Solve $\bar{u}_t(x)$ from $\bar{M}\bar{u}_t(x) = C$, using Thomas algorithm.
6:     **return** $\bar{u}_t(x) \cdot \widehat{(x_1 - x_0)}$. ▷ Magnitude·Direction
7: **end function**

8: **function** GETLATENT($x_0, x_1, t$)
9:     $x_t \leftarrow x_0$.
10:     Uniformly sample $\{0 = t_1 < \cdots < t_{T+1} = t\}$.
11:     **for** $1 \leq i \leq T$ **do** ▷ Euler-step to solve ODE
12:         $v \leftarrow$ GETVELOCITY($x_0, x_1, x_t, t$).
13:         $x_t \leftarrow x_t + (t_{i+1} - t_i) \cdot v$.
14:     **end for**
15:     **return** $x_t$.
16: **end function**

17: **function** GETLOSS($x_0, x_1, x_t, t, h$)
18:     Compute $\bar{M}$ and $C$ according to Eq (11).
19:     $\text{loss}_M \leftarrow ||\bar{M}\bar{u}_\theta(x_t, h) - C||_2^2$.
20:     $\text{loss}_D \leftarrow -\text{cos\_sim}\left(\widehat{u_\theta(x_t, h)}, \widehat{(x_1 - x_0)}\right)$.
21:     **return** $\text{loss}_M + \text{loss}_D$.
22: **end function**

    BEGIN MAIN FUNCTION:
23: Initialize hyper-parameter $\rho = 4.0$.
24: **repeat**
25:     Forward data $\{x_1, h\}$.
26:     Sample $x_0 \sim \mathcal{N}(0,1)$.
27:     Sample $t \sim \mathcal{U}(0,1)$.
28:     $x_t \leftarrow$ GETLATENT($x_0, x_1, t$).
29:     loss$\leftarrow$ GETLOSS($x_0, x_1, x_t, t, h$).
30:     $\theta \leftarrow \arg\min_\theta(\text{loss})$.
31: **until** Converged

---

*Figure 6.* Algorithmic comparison of the training procedures for FM and TFM in text-to-video modeling. The two methods share the same overall training pipeline but differ in the formulation of velocity and latent (*i.e.*, the flow), as well as in the design of the loss function. Notably, the formulation of FM for video generation is identical to that used for image generation, as it does not impose any constraints on temporal relationships between video frames. In contrast, TFM introduces explicit temporal control, making it more suitable for modeling video data.

## B.1. Model and Hyper-parameters

Our base model is built upon Wan2.1-T2V-14B (Wan et al., 2025)[1] , a widely adopted and state-of-the-art open-sourced foundational video generation model. Wan2.1 employs the Diffusion Transformer (DiT) (Peebles & Xie, 2023) architecture, which has increasingly replaced U-Net (Ronneberger et al., 2015) as the dominant backbone for video generation. Given a text prompt, Wan2.1 incorporates the textual information via cross-attention and generates a corresponding video sequence. The model adopts full spatio-temporal attention to effectively capture complex motion and temporal dependencies.

Wan2.1 is trained using standard Flow Matching (FM) on billions of video-text pairs. However, its training objective does not impose explicit constraints on temporal dynamics. We refer readers to their official technical report (`https://arxiv.org/pdf/2503.20314` , *Section 4*) for further details.

Importantly, our proposed TFM is model-agnostic and does not modify the architecture or hyper-parameters of the base video generative model. TFM introduces only one additional hyper-parameter, $\rho$ in Eq (6), which controls the relative strength between the intra-frame and inter-frame terms. In practice, we set $\rho = 4$ to ensure a unique solution and numerical stability of the Thomas algorithm. Larger values of $\rho$ increase the weight of the intra-frame term, thereby weakening temporal constraints; in the limit as $\rho \to \infty$, TFM degenerates to standard FM.

## B.2. Training and Inference

During training, we solve an ordinary differential equation (ODE) to obtain the latent at timestep $t$ (Algorithm 2, line 11). Each ODE step is set to 0.002, resulting in fewer than 500 integration steps for any $t \in [0, 1]$. We evaluate the stability of the ODE integration offline and observe an integration error on the order of $10^{-3}$. We fine-tune the model on the publicly available ShareGPT4Video dataset (Chen et al., 2024)[2] for one epoch, which contains $\mathcal{O}(40k)$ videos paired with text captions annotated by GPT-4V (Achiam et al., 2023). According to the Wan2.1 technical report (`https://arxiv.org/pdf/2503.20314`, *Abstract*), the base model is trained on billions of samples. Therefore, our fine-tuning dataset accounts for roughly $0.01\%$ of the pre-training data, making the fine-tuning stage extremely light-weight.

We fix the number of frames to 81 and the spatial resolution to $832 \times 480$. Fine-tuning is performed using LoRA (Hu et al., 2022) with rank 32. We adopt the AdamW (Loshchilov & Hutter, 2017) optimizer with a learning rate of $1 \times 10^{-4}$ and weight decay of 0.01. Training is conducted on four A100 GPUs with 80GB memory, resulting in a total batch size of 4, and completes in approximately one week.

During inference, we fix the video length and resolution to match the training configuration for both our method and all baselines in Section 5.1. Since CogVideoX1.5 does not support custom output resolutions, its generated videos are spatially resized to the target resolution for fair comparison. We use 50 inference steps and fix the classifier-free guidance scale (Ho & Salimans, 2022) to 5.0. No cherry-picking is performed: all results are generated with a fixed random seed. Inference is carried out on a single A100 GPU with 40GB memory.

# C. Evaluation Details

## C.1. Baselines

We compare our method with the following baselines, which are latest works for text-to-video generation that achieves superior performance. Specifically,

1. **CogVideoX1.5-5B** (Yang et al., 2024): `https://huggingface.co/zai-org/CogVideoX1.5-5B`.

2. **HunyuanVideo-13B** (Wu et al., 2025): `https://huggingface.co/tencent/HunyuanVideo`.

3. **Wan2.1-T2V-14B** (Wan et al., 2025): `https://huggingface.co/Wan-AI/Wan2.1-T2V-14B`.

## C.2. Benchmarks

To comprehensively evaluate the performance of our method in terms of both motion generation and overall video quality, we adopt two publicly available benchmarks, each serving a distinct evaluation purpose:

---

[1]`https://huggingface.co/Wan-AI/Wan2.1-T2V-14B`
[2]`https://huggingface.co/datasets/ShareGPT4Video/ShareGPT4Video`

1. **VideoJam-Bench** (Chefer et al. (2025), `https://arxiv.org/pdf/2502.02492`, *Appendix F*): VideoJam-Bench is specifically designed to assess motion generation capabilities, covering a wide range of dynamic scenarios, including complex human actions and physics-driven motions. The benchmark consists of 128 carefully curated text prompts.

2. **MovieGen Bench** (Polyak et al. (2024), `https://github.com/facebookresearch/MovieGenBench`): MovieGen Bench is a general-purpose video evaluation benchmark that evaluates overall generation quality across diverse content. It contains approximately 1,000 prompts, each annotated with motion intensity levels (*low*, *medium*, and *high*). For computational efficiency, we randomly sample 500 prompts spanning all motion levels.

## C.3. Metrics

The evaluation metrics we adopt can be classified into two categories:

1. **Automatic Metrics**: The automatic metrics are supported by VBench (Huang et al., 2024), spanning six sub-categories, *i.e.*, motion-related as *Motion Smoothness* and *Dynamic Degree*, and appearance-related as *Subject Consistency*, *Background Consistency*, *Asethetic Quality* and *Imaging Quality*. We give the full VBench evaluated results in Table 8 and Table 9. While in the main paper, we aggregate the motion-related and appearance-related metrics respectively, following the procedure specified in the official website of VBench (`https://github.com/Vchitect/VBench`). Concretely, each metric is first normalized then weighted averaged, resulting in the following formulas:

$$
\begin{aligned}
\text{Motion} =& \frac{1}{1.5} \cdot [(\text{Motion\_Smoothness} - 0.706)/(0.9975 - 0.706) + 0.5 \cdot \text{Dynamic\_Degree}]. \\
\text{Appearance} =& \frac{1}{4} \cdot [(\text{Subject\_Consistency} - 0.1462)/(1.0 - 0.1462) + \text{Asethetic\_Quality} \\
& + (\text{Background\_Consistency} - 0.2615)/(1.0 - 0.2615) + \text{Imaging\_Quality}].
\end{aligned}
\tag{46}
$$

Note that in ablation study (Figure 4), we also use the normalized VBench metrics.

2. **User Study**: We conduct a user study following the two-alternative forced-choice (2AFC) protocol. In each trial, participants are presented with two videos (one generated by TFM and the other generated by a baseline method), and are asked to select their preferred video under three criteria: overall quality, motion quality, and semantic coherence. The video pairs are randomly sampled to avoid ordering bias. We recruit 40 participants, each of whom evaluates 10 video pairs. All participants are university students with computer science–related backgrounds, though not necessarily specializing in generative AI. The preference rate of TFM is then computed for each evaluation criterion by aggregating the selections across all participants.

## D. Additional Discussions

### D.1. Formulation of TFM

To motivate the formulation of TFM, we first recall the governing equation (Eq (3)) of standard FM:

$$
d\big(\psi_t(x|x_1^{(i)}), x_1^{(i)}\big) = \tau(t)\, d\big(x_0^{(i)}, x_1^{(i)}\big), \quad 1 \le i \le n,
\tag{47}
$$

which only involves an *intra-frame* term. Consequently, standard FM treats each frame independently and does not explicitly model temporal correlations across frames. To make the flow temporally aware, it is therefore necessary to introduce *inter-frame* terms that impose temporal constraints, which we interpret as motion prior.

The simplest and most natural form of inter-frame interaction is the connection between adjacent frames, *i.e.*, between $\psi_t(x|x_1^{(i-1)})$ and $\psi_t(x|x_1^{(i)})$ for $t \in [0,1]$ and $1 < i \le n$. A minimal way to characterize such an adjacent-frame dependency is through the distance

$$
d\big(\psi_t(x|x_1^{(i-1)}), \psi_t(x|x_1^{(i)})\big).
\tag{48}
$$

A natural extension of Eq (47) is therefore to add this inter-frame term to its left-hand side. However, a naive addition is insufficient, as the flow must still satisfy the prescribed boundary conditions: at $t = 0$, $x_0 = \{x_0^{(1)}, \dots, x_0^{(n)}\}$ should be a valid solution, and at $t = 1$, $x_1 = \{x_1^{(1)}, \dots, x_1^{(n)}\}$ should be the *unique* solution.

Enforcing these boundary conditions determines the corresponding term on the right-hand side of Eq (47), which takes the form of an interpolation between inter-frame distances at $t = 0$ and $t = 1$. This leads to

$$
\begin{aligned}
&\rho d\big(\psi_t(x|x_1^{(i)}), x_1^{(i)}\big) + \mathbb{I}_{i>1}\, d\big(\psi_t(x|x_1^{(i-1)}), \psi_t(x|x_1^{(i)})\big) \\
&= \rho\tau(t)d\big(x_0^{(i)}, x_1^{(i)}\big) + \mathbb{I}_{i>1}\big[\tau(t)\, d\big(x_0^{(i-1)}, x_0^{(i)}\big) + \big(1 - \tau(t)\big)\, d\big(x_1^{(i-1)}, x_1^{(i)}\big)\big],\ 1 \leq i \leq n.
\end{aligned}
\tag{49}
$$

This equation is essentially the governing equation of TFM (Eq (6)), except that in practice, we further extend it by considering not only the connection to the preceding frame, $d\big(\psi_t(x|x_1^{(i-1)}), \psi_t(x|x_1^{(i)})\big)$, but also the connection to the succeeding frame, $d\big(\psi_t(x|x_1^{(i)}), \psi_t(x|x_1^{(i+1)})\big)$. Taken together, these observations indicate that TFM provides a direct and minimal characterization of temporal correlations in video data, while preserving the boundary conditions and structure of flow formulation.

### D.2. When Does TFM Degenerate to Standard FM?

In this section, we examine the relationship between TFM and standard FM. In particular, since we have already shown that TFM models a flow that is conceptually different from standard FM, we aim to characterize the conditions under which TFM degenerates to standard FM.

Recall that the solution for intermediate latent to standard FM is given by $\psi_t(x|x_1^{(i)}) = (1 - \tau(t))x_1^{(i)} + \tau(t)x_0^{(i)}$, for $t \in [0, 1]$ and $1 \leq i \leq n$, which satisfies Eq (47). For any finite $\rho$, substituting this solution into the governing equation of TFM (Eq (6)) yields

$$
\begin{aligned}
&\mathbb{I}_{i>1}d(\psi_t(x|x_1^{(i-1)}), \psi_t(x|x_1^{(i)})) + \mathbb{I}_{i<n}d(\psi_t(x|x_1^{(i)}), \psi_t(x|x_1^{(i+1)})) \\
&= \mathbb{I}_{i>1}[\tau(t)d(x_0^{(i-1)}, x_0^{(i)}) + (1 - \tau(t))d(x_1^{(i-1)}, x_1^{(i)})] + \mathbb{I}_{i<n}[\tau(t)d(x_0^{(i)}, x_0^{(i+1)}) + (1 - \tau(t))d(x_1^{(i)}, x_1^{(i+1)})].
\end{aligned}
\tag{50}
$$

On the other hand, we have

$$
\begin{aligned}
d(\psi_t(x|x_1^{(i-1)}), \psi_t(x|x_1^{(i)})) &= d\big((1 - \tau(t))x_1^{(i-1)} + \tau(t)x_0^{(i-1)}, (1 - \tau(t))x_1^{(i)} + \tau(t)x_0^{(i)}\big) \\
&\geq (1 - \tau(t))d(x_1^{(i-1)}, x_1^{(i)}) + \tau(t)d(x_0^{(i-1)}, x_0^{(i)}),\ 1 < i \leq n.
\end{aligned}
\tag{51}
$$

Combining Eq (51) with Eq (50), we observe that the inequality in Eq (51) must in fact become an equality. This occurs if and only if the two vectors $x_1^{(i)} - x_1^{(i-1)}$ and $x_0^{(i)} - x_0^{(i-1)}$ are collinear and share the same direction. Applying this condition to all $1 \leq i \leq n$, we conclude that TFM degenerates to FM only when, for every frame index $i$, the vectors $x_1^{(i)} - x_1^{(i-1)}$ and $x_0^{(i)} - x_0^{(i-1)}$ are collinear and aligned.

Conversely, consider the scenario in which, for all $1 \leq i \leq n$, the vectors $x_1^{(i)} - x_1^{(i-1)}$ and $x_0^{(i)} - x_0^{(i-1)}$ have the same direction. First, we note that such a condition is almost surely violated in real-world video data. Second, under this highly restrictive scenario, it is straightforward and intuitive that TFM degenerates to FM, since the direction and magnitude of the flow are identical for each frame, and the temporal coupling imposed by TFM introduces no additional constraints beyond those already captured by standard FM.

### D.3. Time Complexity Analysis

In TFM, we introduce additional computations to model a more complex probability path, which is more suitable for capturing temporal dynamics in video data. Importantly, these additional computations are incurred only during training and do not introduce any time or memory overhead at inference. We first analyze the time complexity of TFM from a theoretical perspective. We focus on the GETLATENT function in Algorithm 2, which accounts for the majority of the additional computational cost.

Specifically, the GETLATENT function repeatedly invokes GETVELOCITY within a *for* loop of $T$ iterations. The GETVE-LOCITY function itself consists of two components:

1. *Compute $\bar{M}$ and $C$*: According to Eq (11), the calculation of $\bar{M}$ and $C$ are $\mathcal{O}(nd)$, with $n$ the total number of frames, and $d$ the dimension of latent feature. (Note that $\bar{M}$ is sparse, and we only need to calculate elements along the tri-diagonal)

2. *Solve* $\bar{u}_t(x)$: Using Thomas algorithm (Thomas, 1949), we can efficiently solve the tri-diagonal system in $\mathcal{O}(n)$ time, since $\bar{M} \in \mathbb{R}^{n \times n}$ is diagonally dominant.

Consequently, the overall time complexity of the proposed TFM is $\mathcal{O}(Tnd + Tn)$, compared to $\mathcal{O}(nd)$ for FM. The additional computational cost mainly arises from the iterative loops, which are necessary because the derived ODE does not admit a closed-form solution. We further note that, except for special cases such as the straight flow used in standard FM, most non-trivial flow formulations lack analytical solutions and therefore require numerical ODE solvers. In Section 5.3, we quantitatively show that TFM incurs only a modest additional training cost compared to standard FM, while introducing no extra computational overhead during inference.

### D.4. Decomposition of Velocity in Loss Function

In Section 3.3, we introduce a residual-type loss $\text{loss}^{\text{TFM}}$, in contrast to the solution-type loss $\text{loss}^{\text{FM}}$ used in standard FM. A key design choice underlying this formulation is the decomposition of the target velocity into two components: direction and magnitude. This decomposition is directly motivated by the structure of TFM and the properties of its governing equations.

First, in the TFM formulation, the direction of each frame-wise velocity is fixed by construction, *i.e.*, it is aligned with $x_1^{(i)} - x_0^{(i)}$, while only the magnitudes are coupled through the ODE system. Decomposing the velocity accordingly and assigning separate loss terms to direction and magnitude therefore mirrors the intrinsic structure of the dynamics.

Second, this separation improves optimization stability. Direction prediction is naturally formulated as a normalized similarity objective, which is insensitive to scale, whereas magnitude prediction focuses exclusively on satisfying the linear constraints imposed by the TFM equations, without being confounded by angular errors. Designing distinct losses for direction and magnitude thus leads to a more stable and well-conditioned optimization process, and empirically facilitates effective training of TFM. In ablation study (Figure 4, Table 10 and Table 11), we quantitatively compare the performance of model under different choices of loss functions.

### D.5. Selection of $\rho$

In our formulation, the hyper-parameter $\rho$ controls the relative strength between the intra-frame and inter-frame terms. We theoretically establish that TFM is well-defined, *i.e.*, the flow reaches the clean sample $x_1$ at $t = 1$, when $\rho \geq 4$. In practice, we set $\rho = 4.0$ to ensure a unique solution and maintain numerical stability.

*Table 4.* Mean absolute error of velocities between the TFM flow and FM flow with respect to different values of $\rho$.

| **Value of** $\rho$ | 4 | 5 | 6 | 8 | 10 | 100 |
|---|---|---|---|---|---|---|
| **MAE** | 41.5 | 35.0 | 30.4 | 24.0 | 16.8 | 2.2 |

However, increasing $\rho$ places greater emphasis on the intra-frame term, thereby weakening the temporal constraints; in the limit as $\rho \to \infty$, TFM degenerates to standard FM.

To further illustrate this behavior, we conduct an additional experiment (shown in Table 4) in which we measure the mean absolute error between the per-frame velocity derived by TFM (with different values of $\rho$) and the constant velocity in standard FM. As $\rho$ increases, this difference decreases monotonically, indicating that the flow gradually approaches the independent formulation of standard FM, which does not explicitly capture temporal dynamics.

## E. Additional Experiments

### E.1. TFM across Model Scales and Architectures

In the main experiments presented in Section 5.1, we apply TFM to the Wan2.1-T2V-14B model, one of the largest and widely used open-source text-to-video models in the community. However, our method is agnostic to both model size and architecture. To demonstrate this, we conduct an additional experiment by applying TFM to the substantially smaller Wan2.1-T2V-1.3B model and a different backbone model HunyuanVideo-13B. The experimental settings are the same as in the main experiment.

Specifically, we quantitatively compare the two base models with their TFM-finetuned counterparts on two benchmarks using VBench automatic metrics. The results are summarized in Table 5 and Table 6. As shown, TFM consistently outperforms the corresponding base models across both benchmarks, with particularly pronounced improvements on motion-related metrics. For instance, on VideoJam-Bench, the overall motion score increases from 0.829 to 0.874 for Wan2.1-T2V-1.3B after

*Table 5.* Quantitative comparison between TFM and its base model **Wan2.1-T2V-1.3B** on the two benchmarks, using VBench metrics.

| Benchmark | Method | Motion | | | Appearance | | | | |
|---|---|---|---|---|---|---|---|---|---|
| | | Motion Smoothness | Dynamic Degree | *Overall.* | Subject Consistency | Background Consistency | Asethetic Quality | Imaging Quality | *Overall.* |
| **VideoJam-Bench** | Wan2.1-1.3B | 0.989(±0.008) | 0.547(±0.498) | 0.829 | 0.945(±0.041) | 0.947(±0.029) | **0.570**(±0.077) | 0.606(±0.093) | 0.759 |
| | TFM-1.3B | **0.993**(±0.005) | **0.653**(±0.498) | **0.874** | **0.958**(±0.035) | **0.952**(±0.025) | 0.544(±0.071) | **0.646**(±0.092) | **0.768** |
| **Movie-Gen** | Wan2.1-1.3B | 0.991(±0.008) | 0.460(±0.439) | 0.805 | 0.963(±0.042) | 0.961(±0.028) | 0.574(±0.079) | 0.659(±0.120) | 0.784 |
| | TFM-1.3B | **0.994**(±0.007) | **0.490**(±0.392) | **0.821** | **0.974**(±0.036) | **0.964**(±0.026) | **0.587**(±0.087) | **0.666**(±0.112) | **0.793** |

*Table 6.* Quantitative comparison between TFM and its base model **HunyuanVideo** on the two benchmarks, using VBench metrics.

| Benchmark | Method | Motion | | | Appearance | | | | |
|---|---|---|---|---|---|---|---|---|---|
| | | Motion Smoothness | Dynamic Degree | *Overall.* | Subject Consistency | Background Consistency | Asethetic Quality | Imaging Quality | *Overall.* |
| **VideoJam-Bench** | HunyuanVideo | 0.986(±0.015) | 0.773(±0.419) | 0.898 | 0.931(±0.053) | **0.945**(±0.027) | 0.570(±0.072) | 0.595(±0.110) | 0.752 |
| | TFM-Hunyuan | **0.987**(±0.012) | **0.843**(±0.498) | **0.924** | **0.943**(±0.086) | 0.931(±0.040) | **0.577**(±0.093) | **0.609**(±0.129) | **0.756** |
| **Movie-Gen** | HunyuanVideo | 0.990(±0.009) | 0.470(±0.499) | 0.806 | 0.941(±0.059) | 0.952(±0.032) | **0.596**(±0.082) | 0.642(±0.108) | 0.775 |
| | TFM-Hunyuan | **0.991**(±0.016) | **0.575**(±0.396) | **0.843** | **0.954**(±0.053) | **0.978**(±0.032) | 0.584(±0.087) | **0.648**(±0.125) | **0.787** |

applying TFM, and from 0.898 to 0.924 for HunyuanVideo, corresponding to relative gains of 5.4% and 2.9%, respectively. Meanwhile, the TFM-finetuned models maintain competitive or even superior overall appearance scores.

These results demonstrate the effectiveness, practicality, and scalability of the proposed TFM across models of varying scales, *i.e.*, from compact to large-scale, as well as across different model architectures. Nevertheless, we note that TFM introduces a more challenging training objective than the direct flow formulation used in standard FM. As a result, higher-capacity models are generally better positioned to fully exploit the benefits of TFM and to accurately learn flow representations that are more suitable for video data.

### E.2. Generality beyond Video Generation

Given the inherently temporal nature of TFM, we discuss in this section its potential extension to other temporal domains, such as audio. Specifically, we conduct a preliminary experiment by applying TFM to TangoFlux (Hung et al., 2026), a recent text-to-audio generation framework built upon the standard FM pipeline, which enables a straightforward integration of TFM. We train TangoFlux with both FM and TFM on 50k audio samples from the WavCaps (Mei et al.,

*Table 7.* Quantitative comparison of TFM and standard FM on text-to-audio generation.

| | FD($\downarrow$) | CLAP($\uparrow$) |
|---|---|---|
| **TangoFlux-FM** | 75.88 | 0.202 |
| **TangoFlux-TFM** | **68.24** | **0.226** |

2024) dataset under identical settings, and observe consistent improvements with TFM in terms of FD (Cramer et al., 2019) and CLAP score on the validation set (Table 7). Although this experiment remains preliminary, the results provide encouraging evidence for the potential cross-domain applicability of our method.

### E.3. Trade off between Sample Quality and Inference Efficiency

In this section, we analyze the possible trade-off between sample quality and inference efficiency for TFM and FM. Specifically, we investigate whether TFM requires more inference steps to generate high-quality videos, given that its probability path is more complex. For efficiency, we conduct this experiment using the 1.3B base model and measure video quality using Fréchet Video Distance (FVD) (Unterthiner et al., 2019).

As shown in Figure 7, FVD generally decreases with more inference steps for both FM and TFM. Importantly, TFM does not exhibit noticeable quality degradation with fewer steps compared to standard FM, demonstrating that it can achieve high-quality video generation efficiently. We argue that, although the flow in TFM is more challenging to learn, its explicit modeling of motion priors aligns well with the underlying video dynamics, enabling effective generation without requiring additional inference steps.

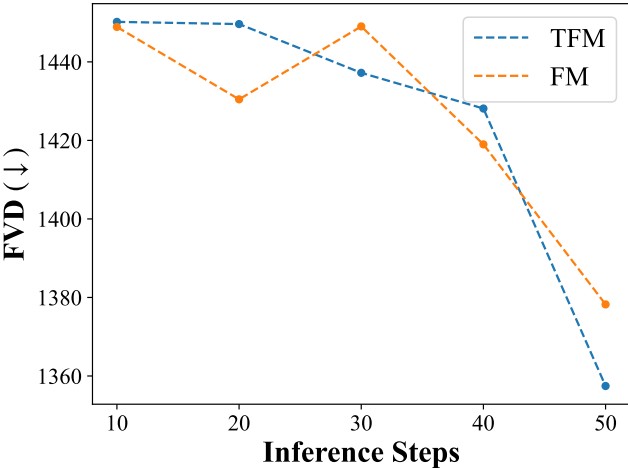

*Figure 7.* Comparison of generation quality in terms of Fréchet Video Distance (FVD) between TFM and FM with respect to the number of inference steps on VideoJam-Bench. FM denotes the base Wan2.1-T2V-1.3B model, while TFM denotes the version fine-tuned with our proposed Temporal-aware Flow Matching, as in Table 5.

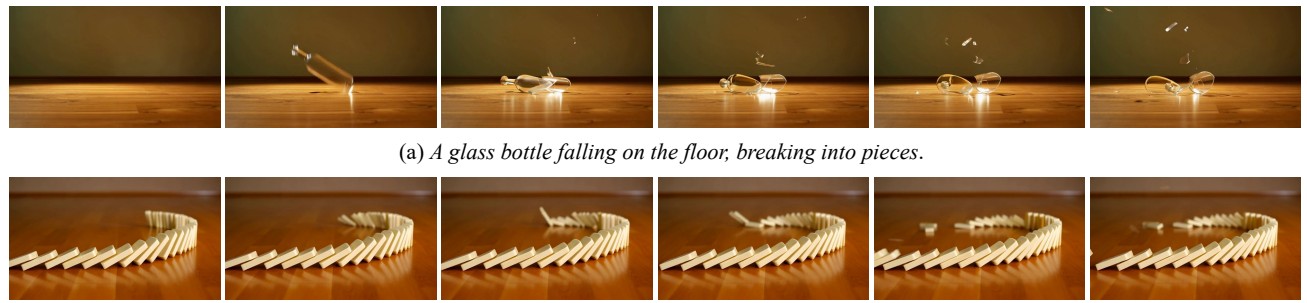

(a) *A glass bottle falling on the floor, breaking into pieces.*

(b) *A domino chain reaction, starting from a single push.*

*Figure 8.* Failed cases. Our method may struggle with highly complex physical processes, including object fracture (*e.g.*, a vase breaking on impact) and chain reactions (*e.g.*, dominoes).

## F. Limitations and Future Work

While TFM provides a more principled theoretical foundation for modeling video data and yields significant improvements in motion generation, it still exhibits several limitations, as illustrated in Figure 8. Empirically, TFM may struggle with highly complex physical processes, such as object fracture (e.g., a vase breaking upon impact) and long-range chain reactions (e.g., domino cascades). Typical failure modes include violations of physical laws and visually ambiguous or incoherent motion patterns, limitations that are also prevalent in most existing models trained with standard FM objectives.

We attribute these shortcomings primarily to the fact that, although TFM is the first method to explicitly introduce temporal constraints into the flow formulation, to the best of our knowledge, it currently adopts only the simplest and most natural mechanism for incorporating temporal control. This leaves substantial room for future explorations, such as modeling richer and more complex probability paths that better capture motion priors inherent in video data. In addition, the fine-tuning data used by TFM is extremely limited, accounting for only approximately 0.01% of the original pre-training data used by the base model. Owing to computational constraints and the lack of large-scale, high-quality public video-text datasets, we are unable to implement TFM under more extensive fine-tuning regimes. Nevertheless, even under this light-weight fine-tuning setting, TFM consistently outperforms existing methods.

Finally, we note that TFM emphasizes intrinsic temporal correlations induced by data structure rather than video-specific assumptions. As a result, the proposed formulation is not restricted to the video domain and can be naturally extended to other forms of temporally structured data, such as audio. A preliminary study on text-to-audio generation is provided in Appendix E.2.

# G. Complete Tables and Additional Figures

In this section, we provide additional tables and figures that complement the results presented in the main paper. Specifically,

- Table 8: A detailed breakdown of the VBench metrics on VideoJam-Bench, including standard errors, corresponding to Table 1 in the main paper.

- Table 9: A detailed breakdown of the VBench metrics on Movie-Gen, including standard errors, corresponding to Table 2 in the main paper.

- Table 10 and Table 11: The exact numerical statistics underlying the ablation study in Figure 4, reported in Table 10 for VideoJam-Bench and Table 11 for Movie-Gen.

- Figure 9: Additional qualitative comparison between TFM and baseline methods.

- Figure 10: Additional generated samples of TFM.

Furthermore, we provide generated video samples in the supplementary material. For ease of evaluation, we have created a webpage (`https://pzrain.github.io/tfm`) showcasing the generated samples and qualitative comparisons with baseline methods.

*Table 8.* Breakdown of VBench metrics with standard errors corresponding to the quantitative comparison between TFM and baseline methods on VideoJam-Bench in Table 1. The top and second top performances have been bolded or underlined respectively.

| Method | Motion | | | Appearance | | | | |
|---|---|---|---|---|---|---|---|---|
| | Motion Smoothness | Dynamic Degree | *Overall.* | Subject Consistency | Background Consistency | Asethetic Quality | Imaging Quality | *Overall.* |
| CogVideoX | 0.974(±0.021) | 0.844(±0.363) | 0.894 | 0.924(±0.044) | 0.952(±0.023) | 0.536(±0.078) | 0.620(±0.113) | 0.750 |
| Hunyuan | 0.986(±0.015) | 0.773(±0.419) | 0.898 | 0.931(±0.053) | 0.945(±0.027) | 0.570(±0.072) | 0.595(±0.110) | 0.752 |
| Wan2.1 | 0.990(±0.008) | 0.716(±0.500) | 0.888 | 0.953(±0.029) | 0.955(±0.022) | **0.591**(±0.071) | 0.648(±0.097) | 0.780 |
| TFM | **0.991**(±0.009) | **0.930**(±0.495) | **0.961** | **0.958**(±0.032) | **0.956**(±0.021) | 0.590(±0.074) | **0.653**(±0.097) | **0.783** |

*Table 9.* Breakdown of VBench metrics with standard errors corresponding to the quantitative comparison between TFM and baseline methods on Movie-Gen in Table 2. The top and second top performances have been bolded or underlined respectively.

| Method | Motion | | | Appearance | | | | |
|---|---|---|---|---|---|---|---|---|
| | Motion Smoothness | Dynamic Degree | *Overall.* | Subject Consistency | Background Consistency | Asethetic Quality | Imaging Quality | *Overall.* |
| CogVideoX1.5 | 0.982(±0.020) | 0.510(±0.500) | 0.801 | 0.949(±0.042) | 0.962(±0.020) | 0.572(±0.094) | **0.673**(±0.114) | 0.783 |
| Hunyuan | 0.990(±0.009) | 0.470(±0.499) | 0.806 | 0.941(±0.059) | 0.952(±0.032) | 0.596(±0.082) | 0.642(±0.108) | 0.775 |
| Wan2.1 | 0.994(±0.006) | 0.440(±0.427) | 0.805 | 0.968(±0.045) | 0.964(±0.025) | **0.601**(±0.086) | 0.649(±0.112) | 0.791 |
| TFM | **0.995**(±0.004) | **0.590**(±0.392) | **0.858** | **0.976**(±0.032) | **0.968**(±0.020) | 0.592(±0.087) | 0.662(±0.111) | **0.795** |

*Table 10.* Ablation study comparing TFM and two variants on VideoJam-Bench, using VBench metrics.

| Method | Motion | | | Appearance | | | | |
|---|---|---|---|---|---|---|---|---|
| | Motion Smoothness | Dynamic Degree | *Overall.* | Subject Consistency | Background Consistency | Asethetic Quality | Imaging Quality | *Overall.* |
| loss$^{FM}$ | 0.983(±0.005) | 0.784(±0.500) | 0.894 | 0.954(±0.036) | 0.952(±0.022) | 0.508(±0.073) | 0.622(±0.109) | 0.752 |
| FM-flow | 0.975(±0.005) | 0.600(±0.400) | 0.815 | 0.942(±0.038) | 0.947(±0.022) | 0.490(±0.101) | 0.609(±0.140) | 0.739 |
| loss$^{TFM}$+TFM-flow (TFM) | **0.991**(±0.009) | **0.930**(±0.495) | **0.961** | **0.958**(±0.032) | **0.956**(±0.021) | **0.590**(±0.074) | **0.653**(±0.097) | **0.783** |

*Table 11.* Ablation study comparing TFM and two variants on Movie-Gen, using VBench metrics, corresponding to Figure 4.

| Method | Motion | | | Appearance | | | | |
|---|---|---|---|---|---|---|---|---|
| | Motion Smoothness | Dynamic Degree | *Overall.* | Subject Consistency | Background Consistency | Asethetic Quality | Imaging Quality | *Overall.* |
| loss$^{FM}$ | 0.994(± 0.003) | 0.460(± 0.367) | 0.811 | 0.974(± 0.034) | 0.954(± 0.034) | 0.504(± 0.097) | 0.633(± 0.135) | 0.756 |
| FM-flow | 0.993(± 0.005) | 0.461(± 0.498) | 0.810 | 0.952(± 0.036) | 0.953(± 0.023) | 0.538(± 0.079) | 0.617(± 0.102) | 0.751 |
| loss$^{TFM}$+TFM-flow (TFM) | **0.995**(± 0.004) | **0.590**(± 0.392) | **0.857** | **0.976**(± 0.032) | **0.968**(± 0.020) | **0.592**(± 0.087) | **0.662**(± 0.111) | **0.795** |

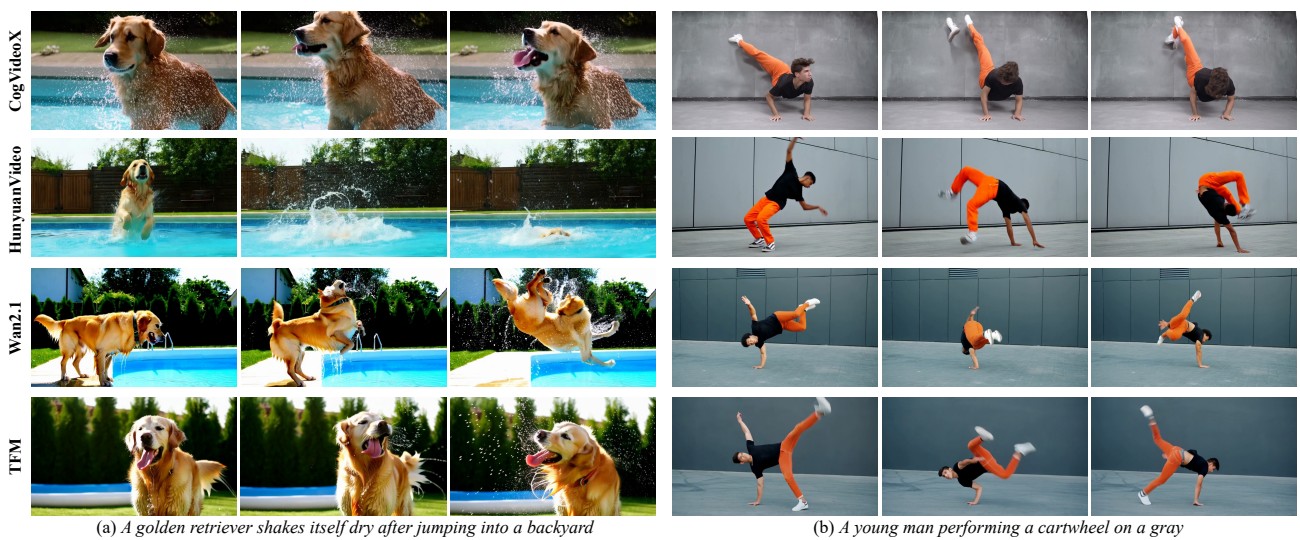

(a) *A golden retriever shakes itself dry after jumping into a backyard pool, droplets spraying out like bright, shimmering stars.*

(b) *A young man performing a cartwheel on a gray surface. He is dressed in orange pants, a black t-shirt.*

*Figure 9.* Additional qualitative comparisons between TFM and baseline methods.

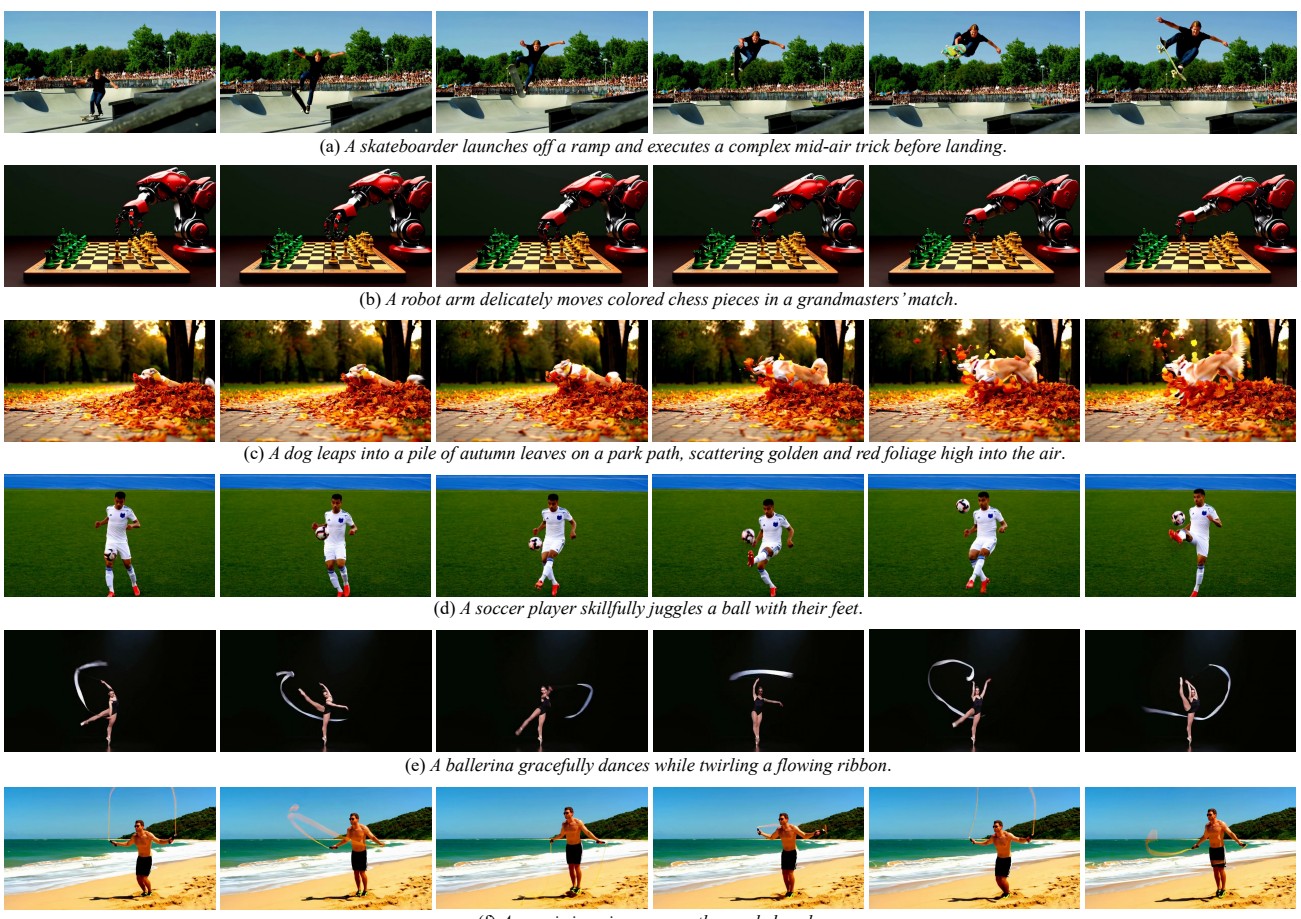

(a) *A skateboarder launches off a ramp and executes a complex mid-air trick before landing.*

(b) *A robot arm delicately moves colored chess pieces in a grandmasters' match.*

(c) *A dog leaps into a pile of autumn leaves on a park path, scattering golden and red foliage high into the air.*

(d) *A soccer player skillfully juggles a ball with their feet.*

(e) *A ballerina gracefully dances while twirling a flowing ribbon.*

(f) *A man is jumping rope on the sandy beach.*

*Figure 10.* Additional generated samples of TFM across various motion types.

