# OpenReview forum: "Temporal-aware Flow Matching for Video Generation with Temporally Coherent Motion"
_ICML.cc/2026/Conference — ICML 2026 regular_

### Official Review · Reviewer_T3Yi · 2026-03-03

**Soundness:** 3
**Presentation:** 2
**Significance:** 3
**Originality:** 2
**Overall Recommendation:** 4
**Confidence:** 4

**Summary:**

This paper proposes Temporal-aware Flow Matching (TFM), a training paradigm that incorporates inter-frame geometric constraints into the flow equations to improve temporal coherence in video generation. The method derives a coupled ODE system with a block tridiagonal coefficient matrix, allowing for efficient velocity field computation and integration into existing flow-based models without inference overhead.

**Compliance With Llm Reviewing Policy:**

Affirmed.

**Final Justification:**

Most of my concerns have been well addressed.

**Key Questions For Authors:**

I find the problem addressed in this paper to be important and the proposed coupled flow system generally logical. Therefore, I lean towards acceptance at the current stage. However, my evaluation depends on the authors addressing the critical concerns regarding experiments and technical claims, especially Weaknesses 1-3.

**Limitations:**

yes

**Strengths And Weaknesses:**

Strengths

1. The paper addresses a critical challenge in video generation with strong motivation.

2. Extending independent flows to a coupled system is a logical and reasonable approach to modeling video dynamics.

3. Proposition 3.1 shows that the coupled flow maintains the property of endpoint consistency.

4. The block tridiagonal structure of matrix M allows for efficient computation of the velocity field.

Weaknesses

1. As noted in Appendix E.1, the proposed TFM is fine-tuned on an additional small dataset while the baselines (e.g., Wan2.1) are original pre-trained models, leading to an unfair comparison in the main results (Tables 1&2). Although Tables 8&9 provide controlled ablations under identical fine-tuning conditions, they do not resolve the unfairness issue.

2. Table 5 shows a significant increase in NetTimeCost for TFM. This suggests that the numerical integration for the coupled ODE system substantially raises the per-iteration training cost. The authors should discuss the training cost in the rebuttal.

3. Why is Wan2.1 used instead of Wan2.2? This raises concerns about whether the reported motion issues have already been resolved in the stronger Wan2.2 baseline, and if the authors are cherry-picking weaker models to exaggerate their method's effectiveness.

4. The user study lacks cross-validation and fails to provide information regarding the participants' backgrounds or professional expertise, undermining the reliability of the subjective evaluation.

5. Since the velocity direction is fixed as a straight line, TFM only modifies the velocity schedule (speed) rather than the geometric path. It is questionable whether such "same path, different speed" design is sufficient to support the ambitious claim of capturing complex physical laws.

6. Although the convergence condition $\rho \ge 4$ is provided, there is insufficient discussion on the numerical stability of the residual loss $\|Mu_{\theta} - C\|^2$. Specifically, the potential for gradient explosion or vanishing during large-scale distributed training remains unaddressed.

---

> ### Author Rebuttal · Authors · 2026-03-30
>
> **W1. Comparison with fine-tuned baselines**
>
> Thank you for the comment. We conduct an additional experiment in which we fine-tune the baselines on the ShareGPT4Video dataset to ensure a fair comparison (see Table 1 on our anonymous website: https://tfm-2026.github.io/). The results show that our method consistently outperforms these fine-tuned baselines across VBench metrics.
>
> **W2. Time complexity**
>
> Thank you for the comment. We would like to clarify that NetTimeCost reflects only the time spent within the algorithm itself (FM/TFM), whereas OverallTimeCost in Table 5 represents the actual per-iteration training time. As shown, on Wan2.1-14B, TFM introduces only a marginal overhead of 2.3% in overall training time, indicating that the practical impact is limited.
>
> As discussed in Appx. E.2, the additional cost mainly arises from the iterative numerical integration of the ODE system. However, this overhead is largely unavoidable for non-trivial flows, which typically do not admit closed-form solutions, except for special cases such as the straight constant flow used in standard FM or certain interpolation curves. We will further revise the related sections to avoid confusions.
>
> **W3. Wan2.2 as baseline**
>
> Thank you for the comment. We did not use Wan2.2 because it was not available when this research began. Wan2.2 was released only six months before ICML submission, and its stable training emerged even later due to the lack of official training scripts. In contrast, Wan2.1 has been widely adopted as a strong and stable backbone.
>
> We also emphasize that our method is model-agnostic. We validate its effectiveness across different backbones, model sizes, and architectures, including Wan2.1-1.3B/14B and HunyuanVideo. Moreover, as stated in Appendix B.2, no cherry-picking is performed in our evaluation.
>
> Following your suggestion, we additionally evaluate Wan2.2 quantitatively and qualitatively (see Table 1 and Figure 2 on our website). While Wan2.2 is generally stronger than Wan2.1, it still exhibits common incoherent issues under complex motions (e.g., body deformations), as illustrated in Figure 2. Quantitatively, although Wan2.2 achieves better imaging quality, which is likely due to its larger-scale pretraining, it still underperforms TFM on motion and consistency metrics, further demonstrating the effectiveness of our method.
>
> **W4. User study**
>
> Thank you for the comment. Our user study follows established prior works [1, 2] using a two-alternative forced-choice (2AFC) protocol. All participants are university students with computer science–related backgrounds, though not necessarily specializing in generative AI. We will further clarify this in the revision. Moreover, the user study results are consistent with objective evaluations (VBench), confirming the improvements achieved by our method.
>
> **W5. Formulation of velocity in TFM**
>
> Thank you for the comment. We clarify that TFM does not simply rescale velocity along a fixed path. Instead, it reformulates the FM governing equation by introducing adjacent-frame constraints, thereby explicitly modeling temporal dynamics. As a result, TFM defines a fundamentally different probability path from the independent flows in standard FM.
> Similar to FM, the probability path in TFM is not unique and depends on the choice of $\tau(t)$, distance metric, and velocity direction. In this work, we adopt a straightforward instantiation, $\tau(t)=1-t$, Euclidean distance, and velocity aligned with $x_1^{(i)} - x_0^{(i)}$, which yields a straight-line trajectory in appearance. However, due to temporal coupling across frames, this path is not equivalent to standard FM.
>
> In Appendix F, we discuss extensions to more flexible probability paths. For example, one may adopt alternative schedules such as $\tau(t)=1-t^2$, or modify velocity directions (e.g., fixing the first frame and solving the remaining tridiagonal system). We will further clarify these in the revised version.
>
> **W6. Numerical stability**
>
> We thank the reviewer for raising concern about potential instability. In our formulation, the matrix $\bar{M}$ (Equ (6)) is tridiagonal, diagonal dominant, and has bounded entries ($|\bar{M}_{i,j}|\leq\rho+2=6$). These properties imply that the operator norm of $\bar{M}$ and $\bar{M}^T$ is bounded, which directly limits the magnitude of the gradient $\nabla_u L=2\bar{M}^T(\bar{M}\bar{u}-C)$ (Equ (12)) and prevents uncontrolled gradient explosion. At the same time, diagonal dominance ensures that the eigenvalues of $\bar{M}^T\bar{M}$ are bounded away from zero, mitigating the risk of gradient vanishing in any direction. Together with the tridiagonal structure, which avoids long-range amplification or decay, this guarantees that graidents remain well-behaved throughout training.
>
> **References**
>
> [1] VideoJAM: Joint Appearance-Motion Representations for Enhanced Motion Generation in Video Models
>
> [2] Stable Video Diffusion: Scaling Latent Video Diffusion Models to Large Datasets

---

> > ### Author Rebuttal · Reviewer_T3Yi · 2026-04-03
> >
> > Thank the authors for their rebuttal. Most of my concerns have been well addressed. I will keep my original positive score.

---

### Official Review · Reviewer_nUHB · 2026-03-11

**Soundness:** 3
**Presentation:** 3
**Significance:** 3
**Originality:** 3
**Overall Recommendation:** 4
**Confidence:** 3

**Summary:**

This paper proposes Temporal-aware Flow Matching (TFM), which modifies the standard flow-matching objective for video generation by coupling adjacent frames in the training flow and replacing the usual solution-matching loss with a residual-style objective, with the aim of improving temporal coherence and motion realism.

**Compliance With Llm Reviewing Policy:**

Affirmed.

**Final Justification:**

I appreciate the authors’ comments, which addressed my concerns. I will maintain my positive score.

**Key Questions For Authors:**

Q1. Generality beyond video generation.
The paper presents TFM as a temporally-aware extension of Flow Matching, which suggests it could apply to general sequential data. However, all experiments are limited to text-to-video generation. Have the authors tested TFM on other types of temporal data (e.g., trajectories, audio, or generic time series)? If not, why should the method be expected to generalize beyond video? Evidence of cross-domain applicability would significantly strengthen the paper’s claims.

Q2. Interpretation of the theoretical analysis.
The theoretical result showing stronger “temporal perception ability” relies on assumptions about frame-difference distributions and a perturbation defined by swapping adjacent frames. Since TFM explicitly couples adjacent frames, the advantage under this setup appears somewhat expected. Can the authors clarify how this theoretical analysis connects to real improvements in motion realism or temporal coherence in practical video generation models?

Q3. Numerical integration details.
The paper states that the TFM trajectories are obtained via numerical integration of an ODE system (using Euler discretization). Could the authors provide more details on the integration procedure (e.g., number of steps and sensitivity to step size)? Since the resulting trajectory defines the training target, it would be helpful to understand how stable the method is with respect to discretization choices.

Q4. Source of performance improvements.
The ablation suggests that some variants produce smoother but less dynamic videos. Could the authors provide further analysis to clarify whether the improvements from TFM come from better temporal modeling, rather than from implicit regularization or motion smoothing effects?

**Limitations:**

yes

**Strengths And Weaknesses:**

Strengths: The paper does not merely append an auxiliary temporal loss, but rewrites the training objective itself by introducing inter-frame constraints into the governing flow equation and deriving a tractable block-tridiagonal system for the induced velocity field. This is more principled than many recent “motion refinement” add-ons. A second positive point is that the method appears operationally lightweight at inference time, since the paper claims the modification affects training only and leaves the standard inference pipeline unchanged.

Weaknesses:

(1) The paper implicitly presents TFM as a more general training paradigm for temporally structured data by modifying the flow-matching objective to incorporate inter-frame temporal coupling. However, the empirical validation is restricted entirely to the text-to-video generation domain. All experiments are conducted on video benchmarks using video generation models, and no experiments are provided on other types of temporal data (e.g., speech, motion trajectories, sensor streams, or generic sequence modeling tasks). As a result, it remains unclear whether the proposed objective is specifically tailored to video frames or whether it genuinely improves learning on broader classes of temporal sequences. This limitation weakens the paper’s broader methodological claims. If the proposed formulation is intended as a general temporal-aware extension of Flow Matching, then at least one experiment on a non-video sequential domain would significantly strengthen the claim of generality. Otherwise, the framing should be adjusted to present the method explicitly as a video-specific training objective rather than a broadly applicable temporal modeling paradigm.

(2) The claimed “enhanced temporal perception ability” rests on assumptions that are almost tautological for natural videos, namely that first-order and second-order frame differences follow different distributions, and then on a highly specific perturbation consisting of swapping two adjacent frames. Under that stylized setup, the result that TFM is more sensitive than FM to the perturbation is not surprising; it is largely baked into the construction because TFM explicitly couples adjacent frames while FM does not. This does not establish a meaningful theory of motion realism, physical plausibility, or even general temporal coherence in modern text-to-video systems. The lower-bound estimate from bootstrap simulation further underscores that the result is an analysis of a toy induced distribution, not of the actual trained generator.

---

> ### Author Rebuttal · Authors · 2026-03-30
>
> **W1 & Q1. Generality beyond video generation**
>
> ||FD($\downarrow$)|CLAP($\uparrow$)|
> |-|-|-|
> |TangoFlux-FM|75.88|0.202|
> |TangoFlux-TFM|**68.24**|**0.226**|
>
> We acknowledge that the main claim of the paper may not be sufficiently clear. We would like to clarify that our goal is not to claim the generality of our method beyond video, but to enhance motion realism in text-to-video generation by introducing explicit inter-frame temporal constraints into the standard FM objective.
>
> That said, given the inherently temporal nature of our formulation, we discuss in Future Work (Appendix F) the possibility of extending our method to other types of temporally structured data, such as audio. We apologize for any confusion and will revise the paper to better reflect this scope.
>
> Moreover, following your suggestion, we conduct a preliminary experiment by applying TFM to TangoFlux [1], a recent text-to-audio generation framework that also adopts the standard FM pipeline. This allows for a straightforward integration of TFM. We train TangoFlux with both FM and TFM on 50k audio samples from the WavCaps dataset under identical settings, and observe consistent performance improvements on the validation set with TFM. While this remains a preliminary study, we believe it provides useful evidence of the potential cross-domain applicability of our method.
>
> **W2 & Q2. Interpretation of the theoretical analysis**
>
> Thank you for this insightful comment. We first clarify that our assumptions are applicable to natural videos. Specifically, the assumptions on first- and second-order distributions generally follow those in Flow Matching, which assumes that all data (e.g., images or videos) come from the same unknown distribution and that the model learns a mapping between a Gaussian distribution and this data distribution.
>
> In Section 4, we study the model’s temporal perception ability—its sensitivity to temporal incoherence—by simulating simple perturbations, where the positions of two consecutive frames are swapped. While this is a basic perturbation, it is generalizable, as any permutation can be decomposed into a sequence of such pairwise swaps.
>
> In Appendix D.4, we further intuitively visualize the temporal perception of a model. For each video, we add noise at a chosen timestep $t \in \{50,100,\dots,900\}$ (out of 1000) and randomly permute the video latents along the temporal axis. We argue that a model with strong temporal perception should show a larger loss increase after permutation, whereas a weaker model would struggle to detect the disruption. In Figure 6, we plot the mean and standard deviation of the relative loss change of both FM and TFM, demonstrating that TFM significantly enhances the model’s ability to detect temporal inconsistencies, and therefore helps improve model realism and temporal coherence in practical video generation. We will further clarify these points in the corresponding sections.
>
> **Q3. Numerical integration details**
>
> Thank you for this insightful comment. To derive the latent $x_t$ at timestep $t \in (0,1)$, we perform iterative ODE integration from $x_0$ with a fixed step size $\Delta t = 0.002$, resulting in a maximum of 500 steps. This choice is motivated by two factors:
>
> 1. **Efficiency**: Our diagonally-dominant, tri-diagonal system enables fast and effective integration. As shown in Appendix E.2 and Table 5, on Wan2.1-14B, TFM adds only **2.3%** overhead during training, with no additional cost at inference.
> 2. **Stability**: We test ODE stability by integrating the full path from $x_0$ to $x_1$ offline, i.e., $x \gets x + \Delta t \cdot u$, and measuring the mean absolute error between the final $x$ and $x_1$. With $\Delta t = 0.002$, the error is on the order of $10^{-3}$–$10^{-4}$, demonstrating numerically stable integration thanks to the favorable properties of our system and the Thomas algorithm.
>
> We will include these details in the revised version.
>
> **Q4. Source of performance improvements**
>
> Thank you for this suggestion. We would like to clarify that our method provides a principled approach to improving motion realism by modifying the flow formulation in standard FM to incorporate inter-frame constraints, explicitly modeling temporal dynamics inherent in video data without any auxiliary regularization. This aligns with Reviewer 9uvY’s observation: “The method directly alters the FM objective without adding any architectural components or any inference-time tricks.”
>
> In our ablation study, TFM significantly outperforms FM under identical settings in Dynamic Degree, reflecting better temporal modeling. In contrast, the other variants show low Dynamic Degree but high Motion Smoothness close to that of TFM, indicating a tendency toward static or under-dynamic motions due to their limited capability in modeling motion dynamics effectively.
>
> **References**
>
> [1] TangoFlux: Super Fast and Faithful Text to Audio Generation with Flow Matching and Clap-Ranked Preference Optimization

---

> > ### Author Rebuttal · Reviewer_nUHB · 2026-04-03
> >
> > Thank you to the authors for the detailed rebuttal and for addressing all of my concerns.

---

### Official Review · Reviewer_9uvY · 2026-03-13

**Soundness:** 3
**Presentation:** 3
**Significance:** 3
**Originality:** 3
**Overall Recommendation:** 5
**Confidence:** 3

**Summary:**

This paper observes motion incoherence in recent flow matching(FM)-based video diffusion models and argues that applying standard FM does not consider temporal dependencies. To address this issue, the authors present Temporal-aware Flow Matching (TFM), which augments the FM equation with adjacent frame temporal constraints, resulting in a tridiagonal system for per-frame velocity magnitudes, while the latent trajectories are estimated via numerical ODE integration during the training and optimized with a residual loss and a cosine direction term. As the proposed method directly changes the training FM objective, it is model-agnostic, and the inference cost remains the same. Authors evaluate this primarily on Wan 2.1 (14B), showing improved motion coherence in VideoJamBench and MovieGen using VBench metrics.

**Compliance With Llm Reviewing Policy:**

Affirmed.

**Final Justification:**

The rebuttal and additional experiments have largely addressed my concerns. I find the work technically solid and a meaningful contribution to video diffusion models. I am therefore raising my overall recommendation from 4 to 5, and encourage the authors to fully incorporate the discussions from the rebuttal period into the final version of the paper.

**Key Questions For Authors:**

- How sensitive are the results to the choice of $\rho$? Did authors consider dynamic sampling or ablation sweep on the value of $\rho$?
- Please refer to the weakness sections for the questions.

**Limitations:**

The paper discusses the potential limitations of the work.

**Strengths And Weaknesses:**

**Strength**
- The paper discusses an important problem of motion realism in t2v generation. The method directly alters the FM objective without adding any architectural components or any inference-time tricks. This could be a great post-training method for general FM-based video models.

- The method is original and theoretically justified. While I have not deeply investigated all the assumptions/proofs, the central algorithm makes sense and the core idea of injecting adjacent-frame temporal constraints into the probability path is meaningfully different from most prior works.

- The paper is easy to follow and presentation is clear.

**Weaknesses**

- I am not sure whether saying base FMs or existing T2Vs **fail** to respect motion priors or temporal dependencies. Despite not explicitly modeling their temporal dependencies, the architectural design and training algorithm allows recent large T2V models to produce physically consistent videos with scaling amount of compute and data. While they might not be extremely accurate in conveying physical laws, I think TFM usually adds marginal or slight physical details and overall big movements are already well modelled via standard FM. I think the word “fail” is too strong.

- The experiment is only conducted on already well pre-trained FM models. TFM’s dynamics from scratch pre-training is not discussed. While the work would still be valuable as a post-training method, readers might be curious about scratch training behaviours. Training TFM with smaller scale video datasets will also be helpful to readers.

- Authors additionally train Wan 2.1 with TFM on ShareGPT4Video data. I would also recommend authors adding Wan 2.1 trained with ShareGPT4Video data with standard FM for fairer comparisons.

---

> ### Author Rebuttal · Authors · 2026-03-30
>
> **W1. Claims on motion modeling**
>
> Thank you for your thoughtful comment. We acknowledge that our method, which is designed to better capture motion priors and preserve physical consistency, is experimentally built upon large pre-trained generative models such as Wan. However, our goal is not to diminish the effectiveness or contribution of Flow Matching or existing T2V approaches. Rather, we aim to provide a novel and principled perspective by introducing inter-frame constraints into the flow formulation, thereby explicitly modeling motion priors inherent in video data. From both theoretical and empirical perspectives, we demonstrate that this leads to improved motion consistency.
>
> We also agree that the wording “fail to capture motion priors” may potentially cause confusion. We will revise it to “fail to explicitly capture motion priors” and carefully update the wording throughout the paper. Thank you again for your valuable suggestion.
>
> **W2. Train from scratch**
>
> Thank you for the comment. Due to computational constraints and the limited availability of large-scale, high-quality video–text datasets, we are unable to train the base model (Wan2.1-T2V-14B) from scratch.
>
> Nevertheless, following your suggestion, we conduct a preliminary study using the smaller Wan2.1-T2V-1.3B model trained from scratch on the ShareGPT4Video dataset with both FM and TFM. We provide the corresponding loss curves on our anonymous website (https://tfm-2026.github.io/, see Figure 1). From these results, we observe that neither model converges under this setting. Consistently, the generated samples from both models fail to preserve basic structural coherence.
>
> These findings suggest that, under limited data and scale, training from scratch remains challenging for both FM and TFM. We will include this analysis of training-from-scratch behavior in the revised version to provide additional insights.
>
> **W3. Ablation: Wan2.1 trained with standard FM**
>
> Thank you for the comment. We would like to clarify that we have conducted experiments directly comparing Wan2.1 trained with TFM and FM under the exact same settings (both trained on ShareGPT4Video). Please refer to the comparison between “FM-flow” and “TFM” in Figure 4, Section 5.2 of the main paper (with detailed statistics provided in Tables 8 and 9). The results show that TFM consistently outperforms Wan2.1 trained with FM, validating the effectiveness of the proposed temporal-aware flow formulation.
>
> **Q1. Choice of $\rho$**
>
> | Value of $\rho$                       | 4    | 5    | 6    | 8    | 10   | 100  |
> | ---------------------------- | ---- | ---- | ---- | ---- | ---- | ---- |
> | Mean Absolute Difference | 41.5 | 35.0 | 30.4 | 24.0 | 16.8 | 2.2  |
>
> Thank you for the suggestion. In our formulation, $\rho$ controls the relative strength between the intra-frame and inter-frame terms. We show that TFM is well-defined, i.e., the flow reaches the clean sample $x_1$ at $t=1$, when $\rho \geq 4$. In practice, we set $\rho = 4$ to ensure a unique solution and maintain numerical stability. However, increasing $\rho$ places greater emphasis on the intra-frame term, thereby weakening the temporal constraints; in the limit as $\rho \rightarrow \infty$, TFM degenerates to standard FM.
>
> To further illustrate this behavior, we conduct an additional experiment (shown in the table above) in which we measure the mean absolute difference between the per-frame velocity derived by TFM (with different values of $\rho$) and the constant velocity in standard FM. As $\rho$ increases, this difference decreases monotonically, indicating that the flow gradually approaches the independent formulation of standard FM, which does not explicitly capture temporal dynamics. We will include this analysis in the revised version.

---

> > ### Author Rebuttal · Reviewer_9uvY · 2026-04-03
> >
> > Most of my concerns have been well addressed. I appreciate the authors' additional efforts, especially the training-from-scratch experiment, which would reasonably be difficult to carry out within such a short rebuttal period; even so, the loss curves are still quite promising. I agree with the other reviewers that some additional discussions on numerical stability should be included in the revision, and that the claims regarding physical law should probably be toned down. Nonetheless, I find the core idea technically plausible and believe it is a meaningful contribution to the community. In light of this, I will increase my score from 4 to 5. Great work!

---

> > > ### Author Response · Authors · 2026-04-08
> > >
> > > We sincerely thank the reviewer for the positive and encouraging feedback on our rebuttal. We are glad to hear that you find the core idea technically plausible and a meaningful contribution to the community. We will carefully revise the manuscript accordingly, and we believe these improvements will further strengthen the quality and clarity of our work.
> > >
> > > We also appreciate your note about increasing the score from 4 to 5. As the discussion period is coming to a close, we would be very grateful if you could kindly update the **Overall Recommendation** in the official review to reflect this change.
> > >
> > > Thank you again for your time and thoughtful evaluation of our work.

---

### Decision · Program_Chairs · 2026-04-30

**Decision:**

Accept (regular)

**Comment:**

This paper proposes Temporal-aware Flow Matching (TFM), extending flow matching for video generation by introducing inter-frame temporal constraints to improve motion coherence while preserving standard inference. Reviewers agree the problem is important and the approach is technically meaningful, with a principled formulation and consistent empirical gains.

In the initial reviews, all three reviewers gave moderately positive scores (4) but raised concerns about claim strength, evaluation scope, fairness of comparisons, and training stability/cost. The rebuttal addressed these points by clarifying scope, improving discussion of baselines and efficiency, and toning down claims. Following this, one reviewer (9uvY) raised their score from 4 to 5, while the other two (nUHB and T3Yi) kept their scores at 4 and indicated their concerns were largely resolved.

Overall, the reviewer consensus is positive, with no score decreases and one increase after rebuttal. While some limitations remain in experimental breadth, the technical contribution is solid and useful for video generation.